# Inhibition of PIP4Kγ ameliorates the pathological effects of mutant huntingtin protein

Ismael Al-Ramahi[1,2], Sai Srinivas Panapakkam Giridharan[3], Yu-Chi Chen[4], Samarjit Patnaik[4], Nathaniel Safren[5], Junya Hasegawa[3], Maria de Haro[1,2], Amanda K Wagner Gee[4], Steven A Titus[4], Hyunkyung Jeong[6], Jonathan Clarke[7], Dimitri Krainc[6], Wei Zheng[4], Robin F Irvine[7], Sami Barmada[5], Marc Ferrer[4], Noel Southall[4], Lois S Weisman[3†], Juan Botas[1,2†]*, Juan Jose Marugan[4]*

[1]Jan and Dan Duncan Neurological Research Institute, Texas Children's Hospital, Houston, United States; [2]Baylor College of Medicine, Texas Medical Center, Houston, United States; [3]Department of Cell and Developmental Biology, Life Sciences Institute, University of Michigan, Ann Arbor, United States; [4]Division of Preclinical Innovation, National Center for Advancing Translational Sciences, Rockville, United States; [5]Department of Neurology, University of Michigan, Ann Arbor, United States; [6]The Ken and Ruth Davee Department of Neurology, Feinberg School of Medicine, Northwestern University, Chicago, United States; [7]Department of Pharmacology, University of Cambridge, Cambridge, United Kingdom

*For correspondence:
jbotas@bcm.edu (JB);
maruganj@mail.nih.gov (JJM)

†These authors contributed equally to this work

Competing interests: The authors declare that no competing interests exist.

**Abstract** The discovery of the causative gene for Huntington's disease (HD) has promoted numerous efforts to uncover cellular pathways that lower levels of mutant huntingtin protein (mHtt) and potentially forestall the appearance of HD-related neurological defects. Using a cell-based model of pathogenic huntingtin expression, we identified a class of compounds that protect cells through selective inhibition of a lipid kinase, PIP4Kγ. Pharmacological inhibition or knock-down of PIP4Kγ modulates the equilibrium between phosphatidylinositide (PI) species within the cell and increases basal autophagy, reducing the total amount of mHtt protein in human patient fibroblasts and aggregates in neurons. In two *Drosophila* models of Huntington's disease, genetic knockdown of PIP4K ameliorated neuronal dysfunction and degeneration as assessed using motor performance and retinal degeneration assays respectively. Together, these results suggest that PIP4Kγ is a druggable target whose inhibition enhances productive autophagy and mHtt proteolysis, revealing a useful pharmacological point of intervention for the treatment of Huntington's disease, and potentially for other neurodegenerative disorders.
DOI: https://doi.org/10.7554/eLife.29123.001

## Introduction

Huntington's disease (HD) is an autosomal dominant neurodegenerative disorder with no curative or preventative treatment options. The disease is caused by the expansion of a translated CAG trinucleotide repeat within exon 1 of the huntingtin gene (*HTT*), resulting in a mutant huntingtin (mHtt) protein with an abnormally long N-terminal tract of glutamine residues (*Ross and Tabrizi, 2011*). Individuals with more than 36 to 39 repeats develop the disorder, and the length of the repeat correlates with the age of disease onset (*Walker, 2007*). The poly-glutamine repeat expansion impacts the physical (*Kazantsev et al., 1999*) and physiological (*Hipp et al., 2012*; *Verhoef et al., 2002*;

Fernandez-Estevez et al., 2014) properties of the huntingtin protein, producing aggregates in aged striatal neurons that eventually precipitate to form neuronal inclusion bodies (Miller et al., 2010a). Accumulation of mHtt triggers a variety of insults that lead to striatal degeneration, however, the nature of the specific mHtt species, soluble, oligomeric or aggregate, that triggers neurodegeneration remains unclear (Arrasate et al., 2004; Lajoie and Snapp, 2010). In the last decade, a number of potential therapeutic avenues have been proposed to prevent or attenuate the neurodegeneration induced by mHtt, including examining the effects of mHtt-induced oxidative stress (Wyttenbach et al., 2002; Giuliano et al., 2003; Lu et al., 2014), huntingtin posttranscriptional modifications (Steffan et al., 2004; Greiner and Yang, 2011; Bhat et al., 2014; Pavese et al., 2006), microglia activation (Gusella and MacDonald, 2009), a systematic exploration of coding (Gusella and MacDonald, 2009) and non-coding (Zhang and Friedlander, 2011) DNA, and autophagy (Sarkar et al., 2009; Williams et al., 2008). However, it has been difficult to identify druggable targets that reduce disease progression (Bard et al., 2014). In addition to targeting mHtt-induced downstream pathogenic events, an attractive alternative for developing HD therapies is reducing the levels of mHtt protein, thus addressing pathogenesis at its root. The therapeutic potential of this approach is supported by observations in animal and cellular models of HD (Sarkar et al., 2009; King et al., 2008; Singh et al., 2014; Giorgini, 2011; Yamamoto et al., 2000; Lin and Qin, 2013; Sarkar et al., 2007). Here we present PIP4Kγ as a novel therapeutic target for HD. PIP4Kγ [Phosphatidylinositol-5-phosphate 4-kinase, type II γ] is a lipid kinase expressed by the PIP4K2C gene. The protein is predominantly localized in several tissues, including the brain (Sasaki et al., 2009; Rameh et al., 1997; Clarke et al., 2008; Clarke et al., 2009). Enzymatically, PIP4Kγ phosphorylates phosphatidylinositol-5-phosphate [PI5P] to produce phosphatidylinositol 4,5-bisphosphate [PI(4,5)P2] (Lietha, 2011). The biological function of PIP4Kγ is not completely understood, although recent reports suggest a role in the modulation of vesicle trafficking (Clarke et al., 2008), and mTOR signaling (Mackey et al., 2014). Recently we presented the first selective inhibitor of PIP4Kγ (Clarke et al., 2015). Here we introduce an additional chemotype with striking cell-based activity which prompted us to explore the utility of inhibiting PIP4Kγ in the context of pathologic mHtt expression. We show that inhibiting PIP4Kγ activity modulates productive autophagy, reduces mHtt protein levels in patient fibroblasts, and clears mHtt aggregates in neuronal cell models. Moreover, we show that inhibition of PIP4Kγ rescues mHtt-induced neurodegeneration in two Drosophila HD models.

## Results

### Identification of novel PIP4Kγ inhibitors

NCT-504 (Figure 1A) is an analogue obtained upon medicinal chemistry optimization of a series of 5-phenylthieno[2,3-d]pyrimidine compounds identified in a high-throughput phenotypic screen (Titus, 2010). Expression of GFP-Htt(exon1)-Q103 in PC12 cells produces detergent-resistant GFP-labeled aggregates (Titus et al., 2012). NCT-504 caused a robust reduction of GFP-Htt(exon1)-Q103 levels, as measured by lowered GFP signal (Figure 1B and C). NCT-504 treatment also decreased huntingtin aggregates in HEK293T cells transiently transfected with GFP-Htt(exon1)-Q74 (Figure 1—figure supplement 1). As thienopyrimidines have been associated with kinase activity (Elrazaz et al., 2015) we profiled NCT-504 against a panel of 442 human kinases http://www.discoverx.com/technologies-platforms/competitive-binding-technology/kinomescan-technology-platform. Using a cutoff of >65% inhibition at 10 µM, NCT-504 was active against only a single kinase, PIP4Kγ (Table 1). Similarly, another analogue from the same thienopyrimidine series, ML168 (Titus, 2010), had activity against six kinases in the same panel, but was most potent against PIP4Kγ.

To better characterize the biochemical action of NCT-504, we evaluated its inhibitory activity in several in vitro kinase assays. NCT-504 modulated the activity of PIP4Kγ in the DiscoverX binding assay (https://www.discoverx.com/services/drug-discovery-development-services/kinase-profiling/kinomescan) with a Kd = 354 nM (Figure 1D). Using a reconstituted assay of phosphorylation of the PI5P substrate by full length PIP4Kγ, NCT-504 inhibited enzyme activity with an $IC_{50}$ of 15.8 µM (Figure 1E). Notably, in the absence of PI5P substrate, the compound did not impair the intrinsic ATP-hydrolytic activity of PIP4Kγ (Figure 1F), suggesting that NCT-504 is an allosteric inhibitor of this kinase. This may account for the differences in potency observed in the enzymatic assay vs the

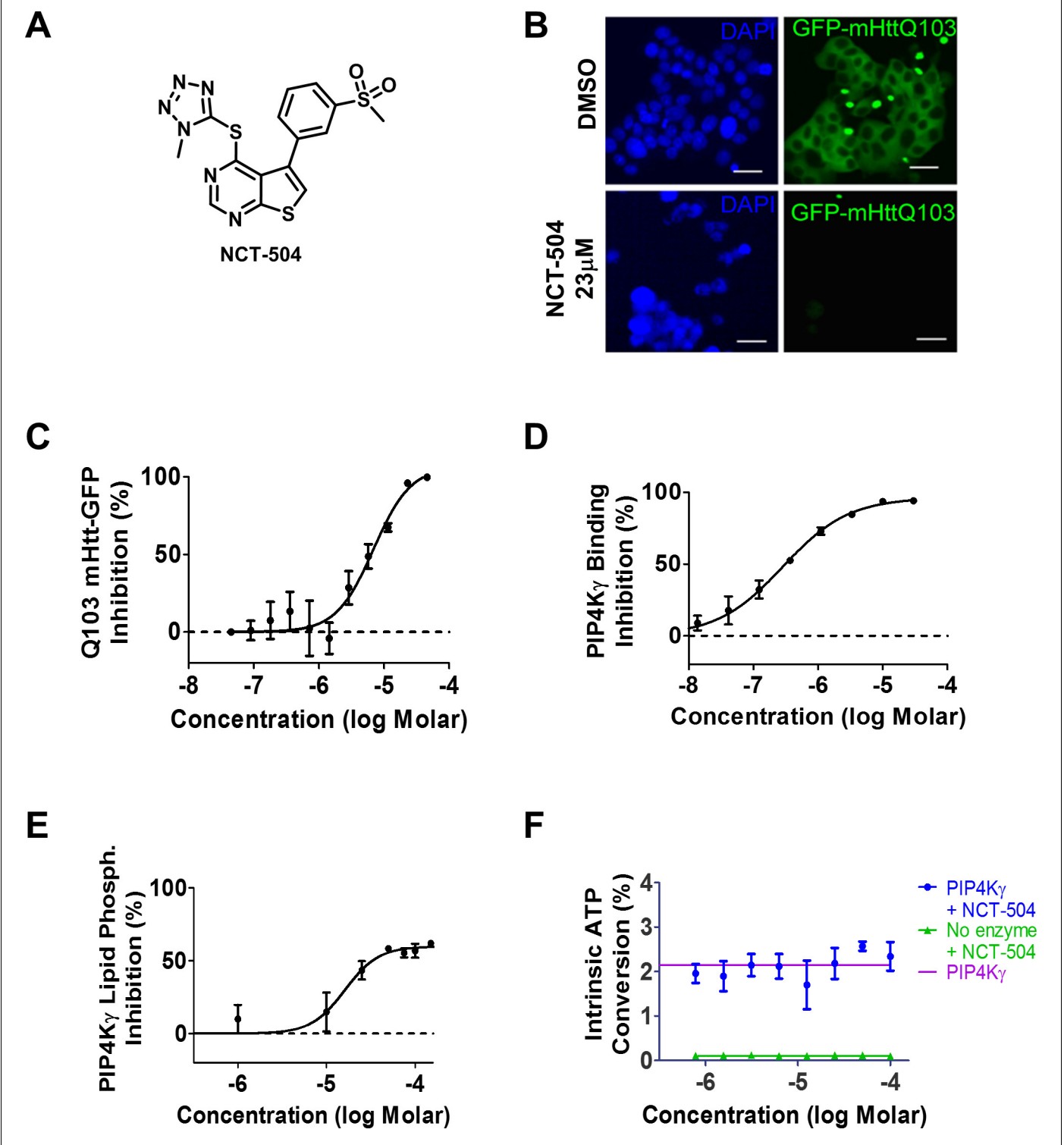

**Figure 1.** Identification of NCT-504 and its inhibition of PIP4Kγ. (**A**) Structure of NCT-504. (**B**) NCT-504 treatment reduces Htt(exon1)-Q103 in PC12 cells. Cells with stable expression of ecdysone-inducible GFP-Htt(exon1)-Q103 (green), induced for 24 hr, and treated with DMSO (top panels) or 23 μM NCT-504 (bottom). Cells stained with DAPI (blue). Scale Bar = 50 μm. (**C**) Concentration-response curve of NCT-504 inhibition of cellular accumulation of GFP-Htt(exon1)-Q103 in PC12 cells. (**D**) NCT-504 inhibition of PIP4Kγ binding to an immobilized proprietary active site ligand (DiscoverX KINOME*scan* https://www.discoverx.com/services/drug-discovery-development-services/kinase-profiling/kinomescan). (**E**) NCT-504 exhibits dose-

*Figure 1 continued on next page*

*Figure 1 continued*

dependent inhibition of phosphorylation of PI4P by full length isolated PIP4Kγ. (**F**) The intrinsic ATPase specific activity of full length isolated PIP4Kγ in the absence of PI5P substrate as a function of NCT-504 concentration is the same in the presence (blue) or in the absence (purple) of NCT-504.

DOI: https://doi.org/10.7554/eLife.29123.002

The following figure supplements are available for figure 1:

**Figure supplement 1.** NCT-504 suppresses the accumulation of HTT-exon1 aggregates.

DOI: https://doi.org/10.7554/eLife.29123.003

**Figure supplement 2.** NCT-504 does not inhibit PIP4Kbeta and weakly inhibits PIP4Kalpha phosphorylation of PI5P.

DOI: https://doi.org/10.7554/eLife.29123.004

**Figure supplement 3.** A PIP4Kγ+ G-loop mutant is resistant to inhibition by NCT-504, consistent with NCT-504 functioning as an allosteric inhibitor.

DOI: https://doi.org/10.7554/eLife.29123.005

DiscoverX binding assay. Similar differences in potency between these two assays have also been observed for allosteric modulators of other kinases (*Rudolf et al., 2014*; *Smyth and Collins, 2009*). NCT-504 function as an allosteric inhibitor may also explain why NCT-504 is exquisitely selective in the kinase profiling assay. In isolated enzyme assays against other PIP4K isoforms, 50 µM NCT-504 did not inhibit PIP4Kbeta or PIP4Kalpha ($IC_{50}$ between 50 µM and 100 µM) (*Figure 1—figure supplement 2*). We also characterized the compound using an alternate PIP4Kγ+ functional assay, which employs PIP4Kγ with a mutated G-loop and two additional mutations (described as PI5P4Kγ G3 + AB in [*Clarke and Irvine, 2013*]) to increase the low intrinsic ATP turnover of the kinase in the presence of PI5P (*Clarke and Irvine, 2013*). NCT-504 was largely inactive against PIP4Kγ+ with a potency >500 µM (*Figure 1—figure supplement 3*).

## PIP4Kγ inhibition modulates cellular phosphatidylinositide levels in complex ways

Cellular inhibition of PIP4Ks should impair the production of PI(4,5)P2 from PI5P, resulting in an elevation of PI5P cellular levels as previously described in the *Drosophila* mutant (*Gupta et al., 2013*). Note that other PI levels were not tested in the dPI4PK *Drosophila* mutant. We hypothesized that elevation of PI5P might further impact the equilibrium between various PI species (*Lietha, 2011*; *Emerling et al., 2014*; *Balla, 2013*). To test this hypothesis, we exposed wild type mouse embryonic fibroblasts to nontoxic concentrations of NCT-504 (10 µM) for 12 hr, and then evaluated the levels of PI by HPLC (*Figure 2*; toxicity assay in *Figure 2—figure supplement 1*). As expected, exposure to NCT-504 elevated cellular levels of PI5P (*Figure 2D*). Surprisingly, NCT-504 also robustly increased PI(3,5)P2 levels, and to a lesser extent increased levels of PI3P (*Figure 2B and E*). We did not observe an effect on PI(4,5)P2 levels (*Figure 2F*), which is consistent with other reports indicating that the cellular levels of this lipid are mostly generated from PI4P via type I PI4P 5-kinases

**Table 1.** Kinase profiling results for NCT-504 and ML168.

Percent activity remaining at 10 µM exposure of NCT-504 and ML168 in KINOMEscan kinase panel/ profiling http://www.discoverx.com/technologies-platforms/competitive-binding-technology/kinomes-can-technology-platform. Top 3 NCT-504 inhibited kinases are reported as single replicate data. Full data set is provided in *Table 1* – source data file. PIP4K2γ potencies were confirmed in triplicate concentration-response testing (*Figure 1D*).

| Kinase | ML168 | NCT-504 |
|---|---|---|
| PIP4K2C | 23 | 4.9 |
| RSK1(Kin.Dom.2-C-terminal) | 20 | 40 |
| GAK | 10 | 42 |

% Control Legend

$0\% \leq x < 10\%$

$10\% \leq x < 35\%$

$35\% \leq x$

DOI: https://doi.org/10.7554/eLife.29123.006

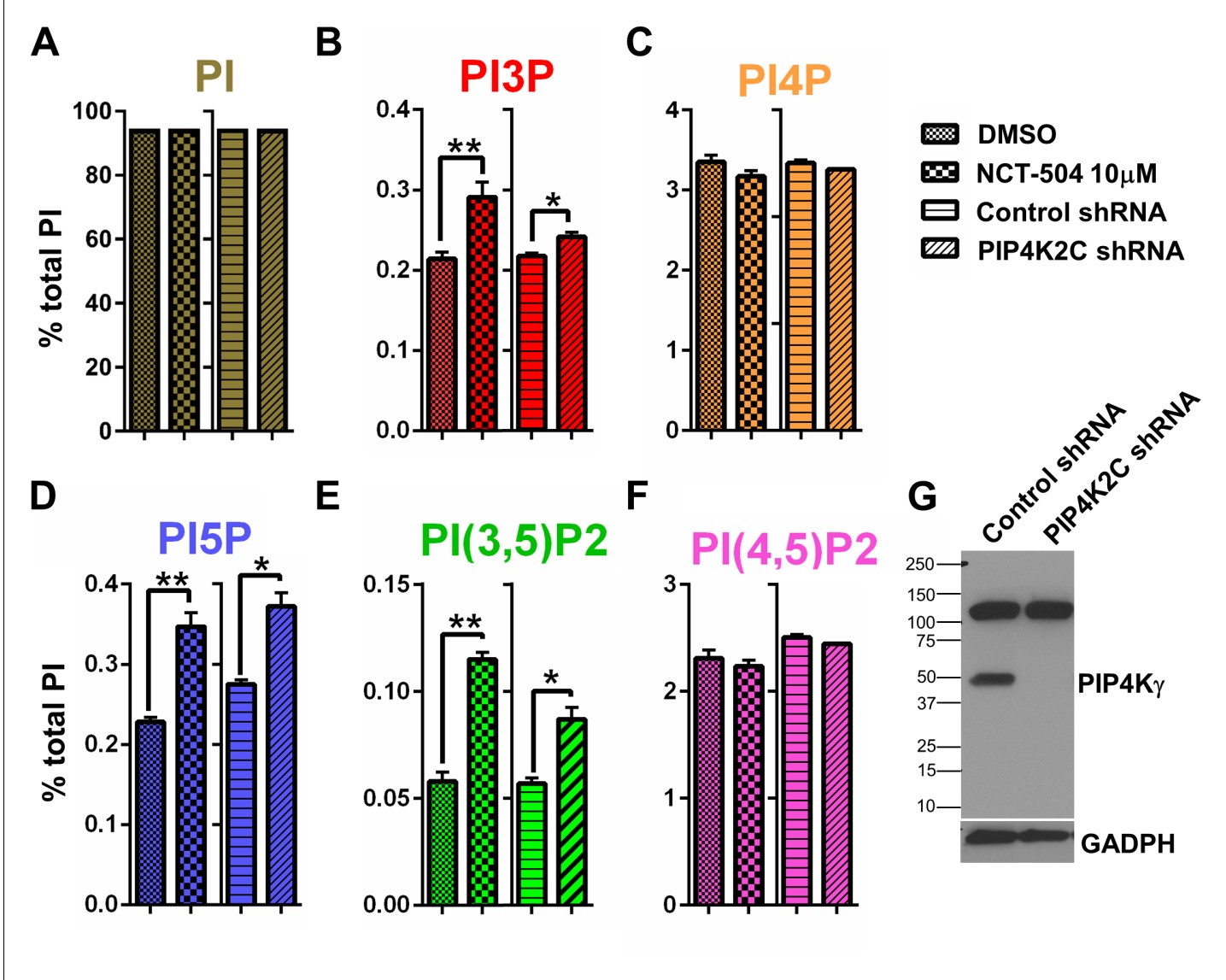

**Figure 2.** Pharmacologic and genetic inhibition of PIP4Kγ elevates the levels of PI(3,5)P2, PI3P and PI5P in MEFs. (A–F) Pharmacologic (NCT-504 10 µM, 12 hr) and genetic (shRNA) inhibition of PIP4Kγ leads to increased levels of PI5P (D), PI(3,5)P2 (E) and PI3P (B), with no significant change in the levels of phosphatidylinositol (A), PI4P (C) or PI(4,5)P2 (F). However, there was a modest reduction in PI4P. Note in *Figure 2—figure supplement 2*, this small change was statically significant. Measurements were performed in MEF cells incubated with $^3$H-inositol labeled media for 48 hr. Statistical significance was analyzed using paired one tailed student t-test (n = 3), *p<0.05, **p<0.01. (G) Anti-PIP4Kγ western blot showing the effective silencing of the enzyme using shRNA. (GAPDH used as loading control).

DOI: https://doi.org/10.7554/eLife.29123.007

The following figure supplements are available for figure 2:

**Figure supplement 1.** NCT-504 treatment does not affect cell viability in MEFs.

DOI: https://doi.org/10.7554/eLife.29123.008

**Figure supplement 2.** Time course of phosphatidylinositol lipid changes upon NCT-504 treatment (10 µM) in MEFs.

DOI: https://doi.org/10.7554/eLife.29123.009

**Figure supplement 3.** Modulation of the levels of phosphatidylinositol lipids by NCT-504 in unaffected human fibroblasts.

DOI: https://doi.org/10.7554/eLife.29123.010

(*Lietha, 2011*). Kinetic measurement of PI levels showed that NCT-504 causes an increase in PI5P, PI (3,5)P2 and PI3P levels along with a decrease in PI4P, progressively over 12 hr (*Figure 2—figure supplement 2*). These statistically significant changes were not observed at 30 or 120 min suggesting that direct inhibition of PIP4Kγ eventually impacts other lipid kinases and phosphatases. Moreover treatment of unaffected human fibroblasts with NCT-504 elevated these three lipids in a dose dependent manner (*Figure 2—figure supplement 3*). Further evidence that the changes in PI levels are due to the specific inhibition of PIP4Kγ, is the finding that shRNA-mediated silencing of PIP4Kγ resulted in a similar PI profile to that observed with NCT-504 inhibition, namely an elevation of PI5P, PI(3,5)P2 and PI3P (*Figure 2B,D and E*). Note that during shRNA-mediated silencing of PIP4Kγ transcripts, PIP4Kγ protein was no longer detected (*Figure 2G*).

## PIP4Kγ inhibition stimulates productive autophagy

Numerous studies have shown that mHtt upregulates autophagy, but impairs incorporation of client proteins into autophagosomes (*Cortes and La Spada, 2014*; *Ochaba et al., 2014*; *Martin et al., 2015*; *Martinez-Vicente et al., 2010*; *Tsvetkov et al., 2013*). Importantly, a number of autophagy modulators have been described that reduce mHtt aggregates (*Sarkar et al., 2009*; *Roscic et al., 2011*; *Zhang et al., 2007*; *Renna et al., 2010*). That NCT-504 elevates the levels of three PI species implicated as positive regulators of autophagy suggests that the observed reduction in HTT-exon1-polyQ aggregates observed with NCT-504 treatment may occur due to upregulation of autophagy. Autophagy can be monitored by following the fate of microtubule-associated protein 1 light chain 3B (LC3-I). During autophagosome formation LC3-I gets conjugated to phosphatidylethanolamine to form LC3-II, which is degraded upon autophagosome-lysosome fusion (*Tanida et al., 2008*). We tested and found that a two hour incubation of HEK293T cells with 5 or 10 μM NCT-504 did not significantly increase LC3-II levels (*Figure 3A and B*). However, LC3-II levels depend on the rate of autophagosome formation, the rate of autophagosome-lysosome fusion, and on the rate of LC3-II degradation in mature autolysosomes. Bafilomycin A1 inhibits the lysosomal v-ATPase, prevents autophagosome-lysosome fusion, and thus prevents autophagy-mediated degradation of LC3-II. Comparison of cells treated with and without bafilomycin A1 is a common method to monitor the rate of autophagosome formation within the cell independent of later steps (*Barth et al., 2010*). Bafilomycin A1 treatment for 2 or 6 hr elevated the total amount of LC3-II (*Figure 3A and C*). Importantly, treating cells with 10 μM NCT-504 and 100 nM bafilomycin A1 for two hours and six hours resulted in a 38% and 51% increase in LC3-II levels respectively compared with bafilomycin A1 treatment alone, which indicates that NCT-504 induces autophagosome formation. Similarly, treating cells with 5 μM NCT-504 and bafilomycin A1 for two and six hours resulted in a 30% and 46% increase in LC3-II levels respectively. Importantly, an elevation in LC3-II levels by NCT-504 in the presence of bafilomycin A1 but not in the absence of bafilomycin A1, suggests that NCT-504 elevates both the induction of autophagy as well as the rate of turnover of autophagic cargo (autophagy flux). To further evaluate the effects of NCT-504 on autophagosome formation and autophagy flux, we used a 293A cell line stably expressing a GFP-mCherry-LC3 reporter (*Figure 3—figure supplement 1*). This double tagged LC3 is commonly used to distinguish between autolysosomes and autophagosomes or phagophores (*Hundeshagen et al., 2011*; *Kimura et al., 2007*). Phagophore and autophagosome membranes conjugated with GFP-mCherry-LC3 are positive for both GFP- and mCherry-fluorescence. Upon generation of mature autolysosomes via fusion of autophagosomes with lysosomes, the GFP fluorescence from the internalized GFP-mCherry-LC3 is quenched in the acidic lysosomes; whereas mCherry fluorescence is insensitive to acidic pH and remains detectable. Thus, membrane structures positive for mCherry fluorescence, but not GFP fluorescence are autolysosomes. We determined the dose and time response of NCT-504 on autophagosomes and autolysosomes using GFP-mCherry-LC3; bafilomycin and torin-1 were used as controls (*Figure 3—figure supplement 1*). As previously reported, bafilomycin treatment resulted in an increase in autophagosomes because the subsequent formation of autolysosomes is blocked. In addition, as previously reported, torin treatment elevated both the number of autophagosomes and autolysosomes, because inhibition of mTORC1 causes an increase in the induction of autophagy as well as an increase in autophagic flux. In contrast, NCT-504 treatment caused a robust increase in the formation of autolysosomes with only a modest elevation in autophagosomes, which indicates that NCT-504 increases autophagic flux, with only a modest increase in autophagy initiation.

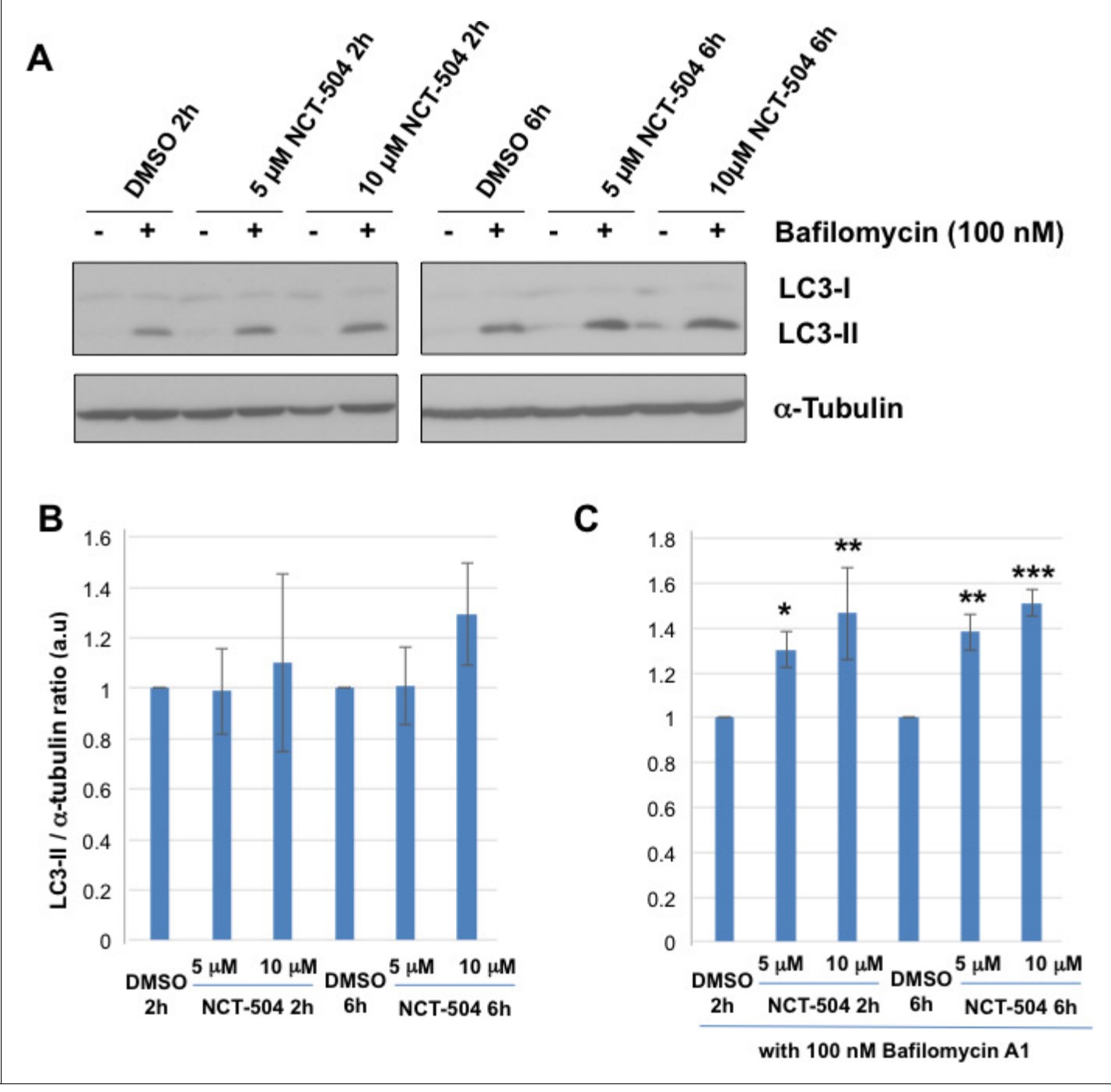

**Figure 3.** Inhibition of PIP4Kγ increases autophagy flux. (A) Representative Western blots showing the levels of LC3-I, LC3-II and Tubulin (loading control) in HEK293T cells treated with either 5 or 10 μM NCT-504 or DMSO (control) for two or six hours in the presence or absence of 100 nM bafilomycin. (B–C) Quantification of LC3-II levels detected by western blot normalized to α-tubulin (loading control). Changes in LC3-II with drug treatment alone is presented relative to levels in the DMSO control cell lysates (B) and changes in LC3-II with drug treatment plus bafilomycin is presented relative to DMSO plus bafilomycin (C). Statistical significance was quantified from three independent experiments using Dunnett's multiple comparisons test, *p<0.05, **p<0.01, ***p<0.005.

DOI: https://doi.org/10.7554/eLife.29123.011

The following figure supplements are available for figure 3:

**Figure supplement 1.** NCT-504 increases autophagy flux and decreases huntingtin protein in 293A cells.

DOI: https://doi.org/10.7554/eLife.29123.012

**Figure supplement 2.** PIP4Kγ inhibition increases the rate of autophagic flux in cortical neurons.

*Figure 3 continued on next page*

*Figure 3 continued*

DOI: https://doi.org/10.7554/eLife.29123.013

**Figure supplement 3.** Lowering mHtt aggregates via NCT-504 requires macroautophagy: (A-B) *Atg7+/+* and *Atg7-/-* cells were transfected with GFP-Htt(exon1)-Q74.

DOI: https://doi.org/10.7554/eLife.29123.014

While mechanisms of autophagy are highly similar in all cells, neurons exhibit some key differences. For example starvation does not upregulate autophagy (*Mizushima et al., 2004*). In addition, autophagy is spatially regulated (*Maday and Holzbaur, 2014*). Thus, we tested whether NCT-504 impacts autophagy in neurons. We tested several doses and time points (up to 72 hr after treatment) and measured autophagy flux in DIV4 rat primary cortical neurons transfected with Dendra2-LC3, a photoconvertible reporter (*Figure 3—figure supplement 2*). Dendra2 has excitation-emission maxima that are similar to GFP. However, exposure to intense blue light raise these maxima, and thus red light is emitted. Since the photoconversion reaction is irreversible, and LC3 is both a marker of autophagy as well as a substrate, the disappearance of red Dendra2-LC3 over time can be used to assess autophagy flux in a noninvasive manner (*Gupta et al., 2017*; *Barmada et al., 2014*). As a positive control for an increase in autophagic flux in neurons, we co-expressed Beclin-1, a positive regulator of autophagy that increases autophagy activity when overexpressed (*Kang et al., 2011*). Importantly, treatment of rat primary cortical neurons expressing Dendra2-LC3 with either 500 nM or 1 µM NCT-504 enhanced the rate of Dendra2-LC3 turnover. Thus, NCT-504 stimulates autophagy flux in primary rodent cortical neurons in a statistically significant manner up to 72 hr following treatment.

That NCT-504-induced changes in autophagic flux were dose dependent, led us to test whether the resultant increase in autophagy correlated with changes in Htt levels. We found that 293A cells display a high content of wt Htt, which enabled us to use an anti-Htt FRET assay (*Cui et al., 2014*). Using this assay, we found that NCT 504 treatment resulted in a dose dependent decrease of Htt protein levels at levels that did not impact cell viability (*Figure 3—figure supplement 1C and D*).

To further test whether NCT-504 reduces mHtt aggregates via increasing autophagic flux, we tested the ability of NCT-504 to lower GFP-Htt(exon1)-Q74 aggregates in a cells with a defect in macroautophagy. We found that while NCT-504 lowered the levels of GFP-Htt(exon1)-Q74 aggregates in Atg7+/+ MEF, it failed to lower aggregates in Atg7-/- MEF; Atg7 is essential for autophagosome formation and its loss inhibits the autophagy pathway (*Figure 3—figure supplement 3*).

That PI3P is a critical regulator of autophagy (*Shibutani and Yoshimori, 2014*), and that PI5P and PI(3,5)P2 have also been implicated in the autophagy process (*Vicinanza et al., 2015*; *Hasegawa et al., 2017*), suggests that upregulation of one or more of these lipids is the driver behind the increase in autophagic flux. Importantly, NCT-504 treatment contrasts with the action of other autophagy modulators such as mTORC1 inhibitors which produce stable increases in LC3-II (*Boland et al., 2008*), accelerating the initiation of autophagy but not necessarily later steps which require mTOR reactivation (*Munson and Ganley, 2015*).

## Blocking PIP4Kγ activity reduces levels of full-length mutant huntingtin protein and levels of Htt(exon1)-polyQ aggregates

To test whether PIP4Kγ inhibition lowers full-length mHtt protein, we used immunoblots to determine the effect of NCT-504 on mHtt levels in patient fibroblasts and immortalized striatal neurons from a knock-in HD mouse model. Notably, treatment with 5 µM NCT-504 for 12 hr, conditions that did not affect cell viability (*Figure 4—figure supplement 1*), significantly reduced mHtt levels in fibroblasts from two different HD patients HD(Q68) or HD(Q45) (*Figure 4A and C*). To further test whether the reduction of mHtt levels was due to selective modulation of PIP4Kγ, we individually silenced PIP4K2A, PIP4K2B and PIP4K2C RNA in the HD(Q68) patient fibroblast cell line. Only silencing of PIP4K2C exhibited an appreciable and robust reduction of huntingtin protein levels (*Figure 4B*). Note that silencing of PIP4K2A, PIP4K2B and PIP4K2C was effective and specific for each isoform (*Figure 4—figure supplement 2*). We also tested the effect of NCT-504 on the levels of mutant full-length huntingtin protein in immortalized striatal neurons. We treated a striatal cell line from a knock-in HD mouse (ST*Hdh*Q111) (*Trettel et al., 2000*), with 5 µM NCT-504 for 12 hr and observed a 40% decrease in mHtt levels (*Figure 4D*).

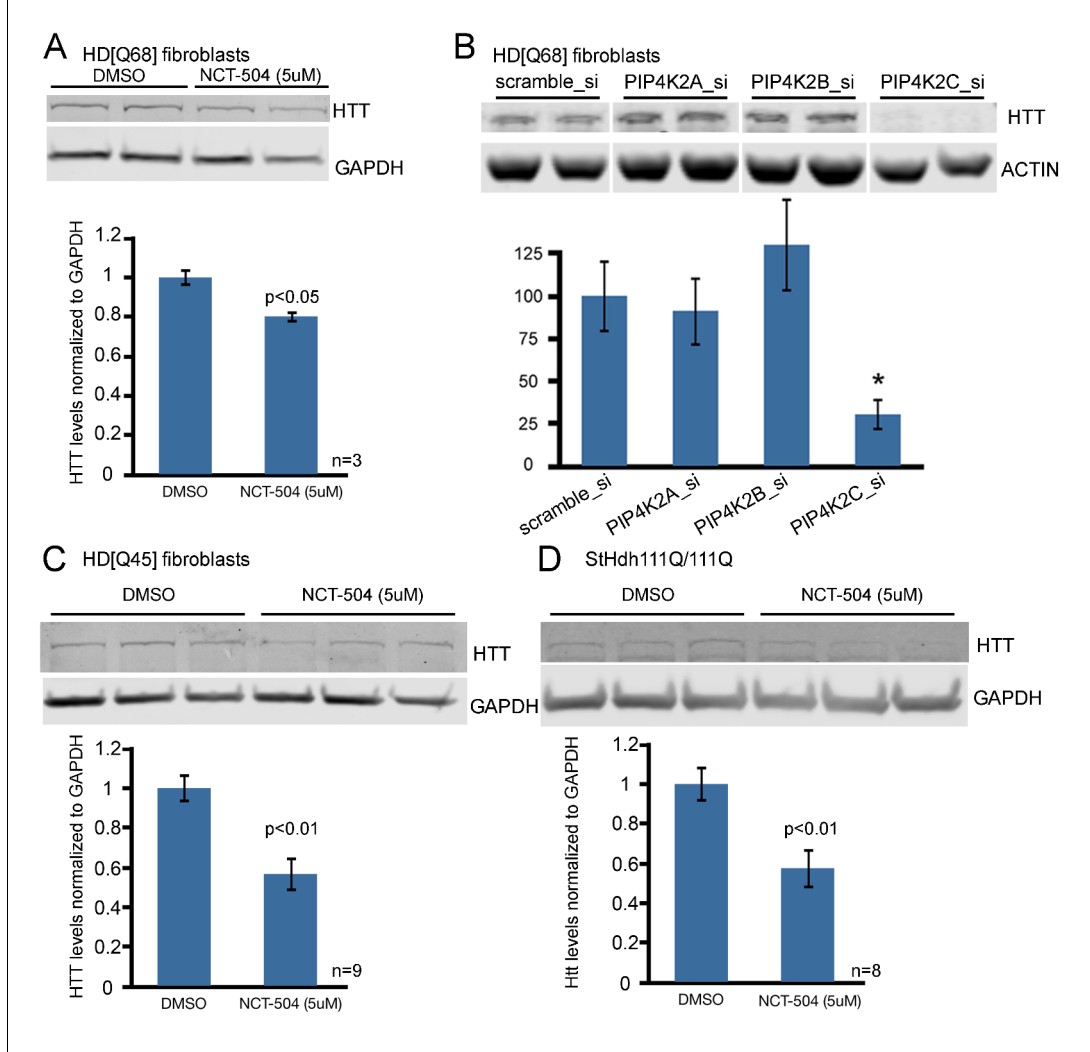

**Figure 4.** Chemical inhibition of PIP4Kγ or knock-down of the corresponding mRNA, PIP4K2C, lowers mHtt protein levels in cells from HD patients and HD knock-in mice. (**A**) Reduction of mHtt protein levels in an HD patient fibroblast cell line (Q68) following exposure for 12 hr to NCT-504 (5 μM) (**B**) mHtt protein levels in patient fibroblast cell line (Q68) were analyzed following siRNA-mediated silencing of PIP4K2A, PIP4K2B and PIP4K2C genes. Note that only PIP4K2C knockdown lowers mHtt levels. Control experiments showing silencing specificity on PIP4K protein levels are in *Figure 4— figure supplement 3*. (**C**) Reduction of mHtt protein levels in an HD patient fibroblast cell line (Q45) following exposure to NCT-504 (5 μM). (**D**) Reduction of mHtt protein levels in immortalized striatal cells from knock-in HD mice (ST*Hdh*Q111) treated for 12 hr with NCT-504 (5 μM).
DOI: https://doi.org/10.7554/eLife.29123.015

The following figure supplements are available for figure 4:

**Figure supplement 1.** Cell viability of HD patient fibroblasts (Q45) exposed to the indicated doses of NCT-504 for 12 hr as per the CellTiter-Glo Promega assay.
DOI: https://doi.org/10.7554/eLife.29123.016

**Figure supplement 2.** Knock-down efficiency and specificity of small interfering RNA in HD patient fibroblasts (Q45).
DOI: https://doi.org/10.7554/eLife.29123.017

**Figure supplement 3.** Experimental details and controls for mouse primary cortical neurons transduced with Htt(exon1)-Q72.
DOI: https://doi.org/10.7554/eLife.29123.018

**Figure supplement 4.** Reduction of Htt protein levels or aggregate by inhibition of PIP4Kγ or PIP4K2C knockdown.
DOI: https://doi.org/10.7554/eLife.29123.019

**Figure supplement 5.** Effect of PIP4K2C knockdown on mHtt aggregates in N2a transfected cells.
DOI: https://doi.org/10.7554/eLife.29123.020

To examine the impact of NCT-504 on the levels of huntingtin-related aggregates in neurons, we evaluated the effect of NCT-504 in wild-type mouse primary cortical neurons transfected with Htt (exon1)-Q74. We tested and found that concentrations of NCT-504 of 5 µM or lower did not impact the viability of cortical neurons (*Figure 4—figure supplement 3*). Importantly, 2.5 or 5 µM NCT-504 lowered the levels of Htt(exon1)-Q74 in primary cortical neurons (*Figure 4—figure supplement 4A*). Moreover, depletion of PIP4Kγ in cortical neurons via PIP4K2C-shRNA treatment also led to a decrease in Htt(exon1)-Q74 levels and Htt(exon1)-Q74 aggregates (*Figure 4—figure supplement 4B*). Furthermore, NCT-504 treatment and PIP4K2C silencing each reduced Htt(exon1)-polyQ aggregates in neuroblastoma N2a cells transfected with Htt(exon1)-polyQ mutants (*Figure 4—figure supplement 5*).

Collectively, these studies show that NCT-504, a PIP4Kγ kinase inhibitor, at non-toxic concentrations, reduced full length huntingtin protein in patient fibroblasts, in immortalized striatal neurons from ST*Hdh*Q111 mutant mice and in HEK293T cells. Moreover, NCT-504 reduced the levels of Htt(exon1)-polyQ aggregates in primary cultured neurons and several cell lines. Similarly, specific silencing of the PIP4K2C gene led to reduction in the levels of full-length huntingtin and HTT-exon1-polyQ protein and aggregates. The lowering of huntingtin and Htt(exon1)-polyQ by NCT-504 was concentration dependent. Moreover, the levels of NCT-504 that reduced these mutant proteins increased autophagic flux. Importantly, NCT-504 did not lower Htt(exon1)-polyQ protein in Atg7-/- MEF, but lowered Htt(exon1)-polyQ protein in the corresponding Atg7+/+ MEF. Together these studies indicate that inhibition of PIP4Kγ lowers mutant Htt, via an increase in autophagic flux.

## Phenotypic effects of PIP4K modulation in *Drosophila* models of Huntington's disease

Unlike mammals, which have three PIP type II enzymes (PIP4Kalpha, PIP4Kbeta and PIP4Kγ), there is only one type II PIP kinase homologue in *Drosophila* (dPIPK4 also called CG17471) (*Mackey et al., 2014*). We used a well-established HD *Drosophila* model (*Kaltenbach et al., 2007*; *Branco et al., 2008*; *Miller et al., 2010b*; *Lu et al., 2013*; *Yao et al., 2015*) to evaluate the impact of modulating the dPIP4K gene on the pathogenesis induced by mHtt expression. The GAL4/UAS system (*Elliott and Brand, 2008*) is used to drive expression of an N-terminal human 128Q mHtt (HttN231Q128) fragment to the cell type of choice. First, we assessed the *Drosophila* retina and its photoreceptor cells. Control HD model animals with wild-type activity of dPIP4K show prominent mHtt-induced photoreceptor degeneration. This phenotype is ameliorated by reducing dPIP4K activity with either one of two different shRNAs (*Figure 5A*). In a second set of experiments we tested the potential of dPIP4K to modulate mHtt pathogenesis using a behavioral readout. Neuronal-specific expression of HttN231Q128 leads to a late-onset motor impairment that can be quantified in a climbing assay. This phenotype was also ameliorated by reducing the activity of dPIP4K using a previously described (*Gupta et al., 2013*) classical loss-of-function mutant allele in heterozygosis and a kinase dead allele (*Figure 5B*). Additionally, we also evaluated these approaches (loss-of-function by a heterozygous mutant allele and kinase dead allele) in animals expressing full length Htt carrying a 200 polyQ expansion in exon1. Notably, we observed a mitigation of the motor performance decline in this full-length HD model. Decreasing the levels of PIP4K with the same alleles in the absence of mHtt did not affect motor performance when compared to controls (*Figure 5—figure supplement 1*). Thus, reducing the activity of dPIP4K using different genetic approaches mitigates mHtt pathogenesis in three different assays.

## Discussion

Our unbiased screen for compounds that protect cells against a pathogenic huntingtin fragment reveal PIP4Kγ as a potential target for Huntington disease. The compounds identified led to the development of NCT-504, a selective fully efficacious inhibitor of PIP4Kγ. NCT-504 treatment or knock-down of PIP4Kγ lowers huntingtin fragments Htt(exon1)-polyQ in multiple cell types including cortical and striatal neurons, and lowers full-length mutant huntingtin in patient fibroblasts and mouse striatal neurons. Moreover, genetic targeting of PIP4K in two Drosophila models of HD, mitigated associated HD phenotypes. Importantly, we observed two major changes in cells following PIP4Kγ inhibition, an increase in autophagic flux, and an increase in the levels of three

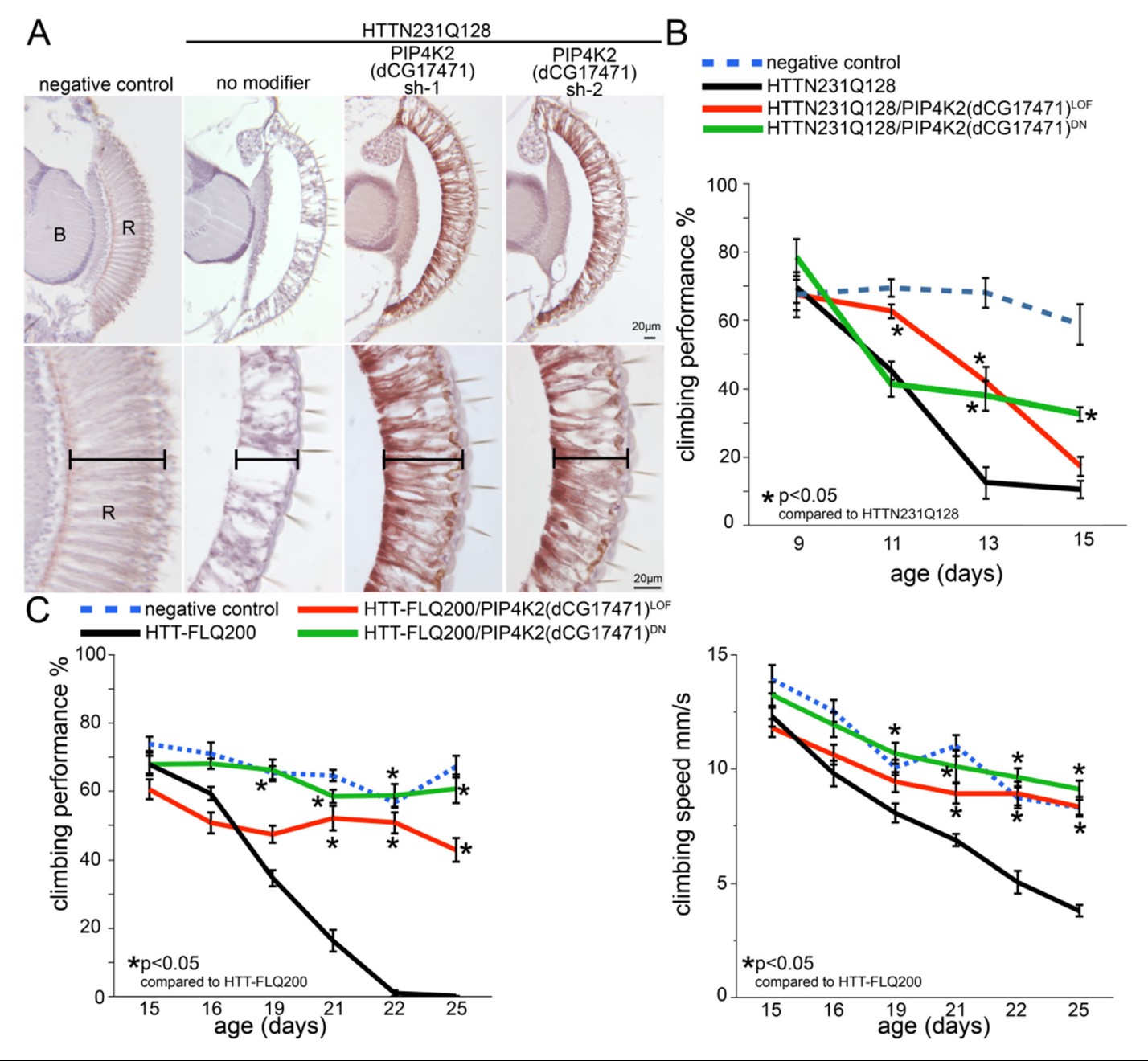

**Figure 5.** Reduced dPIP4K gene activity ameliorates photoreceptor degeneration and behavioral impairments in a Drosophila HD model. (**A**) Sections through the *Drosophila* retina showing loss of photoreceptor cells and retinal tissue in animals expressing N-terminal mHtt (HTTNT231Q128) in the eye (compare no modifier with negative control panels). The photoreceptor and retinal loss phenotype is ameliorated in HttNT231Q128 animals that also express anyone of two shRNAs targeting dPIP4K. (**B**) Chart shows motor performance (%) as a function of age in negative controls (dPIP4K+/+, blue dotted line), *Drosophila* expressing N-terminal mHtt in the CNS (HTTNT231Q128/dPIP4K+/+, black line) or animals expressing N-terminal mHtt in the CNS together with a dPIP4K heterozygous loss of function (HTTNT231Q128/dPIP4K+/-, red continuous line) or a dPIP4K kinase dead isoform (HTTNT231Q128/dPIP4K+/DN, green continuous line). Notice the amelioration of mHtt-induced deficits upon decreasing the activity of dPIP4K. (**C**) Chart shows motor performance (%) and climbing speed as a function of age in negative controls (dPIP4K+/+, blue dotted line), *Drosophila* expressing full length mHtt in the CNS (HTT-FLQ200/dPIP4K+/+, black line) or animals expressing FL mHtt in the CNS together with a dPIP4K heterozygous loss of function (HTT-FL200/dPIP4K+/-, red continuous line) or a dPIP4K kinase dead isoform (HTT-FL200/dPIP4K+/DN, green continuous line). Note amelioration of neural HttNT231Q128-induced motor deficits by decreasing the activity of dPIP4K. Genotypes in A: Negative control: *GMR-GAL4/+; dPIP4K+/+*. No modifier: *GMR-GAL4/+; UAS:HTTNT231Q128/+; dPIP4K+/+*. PIP4K sh1/sh2: *GMR-GAL4/+; UAS:HTTNT231Q128/UAS:dPIP4Ksh-1 or sh-2*. Genotypes in B: Negative control: *elavc155GAL4/+; dPIP4K+/+*.HTT231Q128: *elavc155GAL4/+; UAS:HttNT231Q128/+; dPIP4K+/+*. HTT231Q128/

*Figure 5 continued on next page*

Figure 5 continued

PIP4K2$^{LOF}$: elavc155GAL4/+; UAS:HttNT231Q128/+; dPIP4K29/+ and HTT231Q128/PIP4K2$^{DN}$: elavc155GAL4/+; UAS:HttNT231Q128/UAS:dPIP4K29 [D271K].. Genotypes in C: Negative control: elavc155GAL4/+; dPIP4K+/+. HTT-FLQ200: elavc155GAL4/+; UAS:HttFLQ200/+; dPIP4K+/+. HTT-FLQ200/PIP4K2$^{LOF}$: elavc155GAL4/+; UAS:UAS:HttFLQ200/+; dPIP4K29/+ and HTT-FLQ200/PIP4K2$^{DN}$: elavc155GAL4/+; UAS:UAS:HttFLQ200/UAS:dPIP4K29 [D271K]. elavc155GAL4 drives expression of mHtt to all neurons but not other cell types. Means between points at each age were analyzed by ANOVA followed by Dunnet's post hoc test. Error bars indicate the s.e.m. *p<0.05.

DOI: https://doi.org/10.7554/eLife.29123.021

The following figure supplement is available for figure 5:

**Figure supplement 1.** Reduced dPIP4K gene activity in wild type Drosophila does not impact motility.

DOI: https://doi.org/10.7554/eLife.29123.022

phosphoinositide signaling lipids. It is tempting to speculate that the changes in PI upregulate autophagic flux, and thereby lower mHtt levels.

Little is currently known about cellular roles of PIP4Kγ. However, in line with our current findings, previous studies observed that knock-down of PIP4Kγ resulted in an increase in autophagy (*Mackey et al., 2014*; *Vicinanza et al., 2015*), and a reduction of EGFP-HttQ74 aggregates in MEFs that was dependent on the presence of the autophagy gene, ATG7 (*Mackey et al., 2014*; *Vicinanza et al., 2015*).

While increasing the proteolysis of pathogenic huntingtin protein via the upregulation of autophagy is an attractive therapeutic approach for HD (*Lin and Qin, 2013*; *Sarkar et al., 2007*), there are potential challenges. Mutant huntingtin itself may alter autophagy. Wild-type huntingtin may be an adaptor for selected autophagic cargoes including itself (*Martinez-Vicente et al., 2010*). Consistent with this hypothesis, mutant huntingtin impairs the loading of ubiquitinated-tagged proteins into autophagosomes (*Martinez-Vicente et al., 2010*), and circumvents its own clearance (*Martin et al., 2015*). Moreover, mutant huntingtin sequesters diverse proteins required for key cellular processes (*Kim et al., 2016*), including mTOR, which plays key roles in the regulation of autophagy (*Tsvetkov et al., 2013*; *Petersén A et al., 2001*; *Ravikumar et al., 2004*; *Pryor et al., 2014*; *Ashkenazi et al., 2017*; *Caviston et al., 2007*; *Wong and Holzbaur, 2014*). In addition, the PI binding autophagy adaptor protein ALFY which plays a fundamental role in degrading mutant huntingtin is down-regulated in HD (*Martin et al., 2015*; *Filimonenko et al., 2010*). In contrast with these findings, there are studies that indicate that autophagy is not impaired in HD, and have revealed an elevation of autophagy flux in HD cells (*Petersén A et al., 2001*; *Kegel et al., 2000*). Despite possible mutant huntingtin-dependent changes on autophagy, upregulation of autophagy remains a viable approach for lowering levels of mutant huntintin and aggregates (*Lin and Qin, 2013*; *Sarkar et al., 2007*). Note that caloric restriction also raises the basal level of autophagy, leading to improvements in HD models (*Duan et al., 2003*) and increasing axonal autophagosome transport (*Ikenaka et al., 2013*), although it is less clear how to translate this observation into clinical practice.

One common approach to induce autophagy is via inhibition of the major metabolic kinase mTORC1. Indeed, rapamycin, an inhibitor of mTORC1 also reduces mHtt protein levels (*Sarkar et al., 2009*). However, while PIP4Kγ likely impacts mTORC1 activity, it is not yet clear whether PIP4Kγ inhibition results in mTORC1 inhibition or activation. One study showed that knockdown of PIP4K2C inhibits mTORC1 (*Mackey et al., 2014*). However, in PIP4K2C homozygous knockout mice, mTORC1 is elevated (*Shim et al., 2016*). Thus, the precise link between PIP4Kγ and mTORC1 is not clear. Importantly, our data suggest that PIP4Kγ upregulation of autophagy has some differences with upregulation of autophagy via mTORC1 inhibition. While inhibition of mTOR via torin treatment exhibited a large increase in both autophagosome formation and autophagy flux, inhibition of PIP4Kγ had only a modest impact on autophagosome formation, but had a large increase in autophagic flux (*Figure 3—figure supplement 1*). Thus, elucidation of the mechanism whereby PIP4Kγ inhibition increases autophagic flux remains to be fully determined.

It is likely that inhibition of PIP4Kγ increases autophagic flux at least in part via the resulting impact on the levels of selected phosphoinositide lipids. PIP4Kγ is predicted to convert PI5P to PI(4,5)P2. However, it was not known which cellular pools of PI5P are substrates for PIP4Kγ. Using NCT-504 we found no significant change in PI5P levels or other PI lipids following up to two hours of inhibition of PIP4Kγ. This contrasts with other lipid kinases such as PIKfyve, where direct inhibition results in an acute loss of PI(3,5)P2 which can be observed within 5 min (*Zolov et al., 2012*;

*McCartney et al., 2014*). The long delay prior to changes in PI5P following PIP4Kγ inhibition suggests that PIP4Kγ is not in contact with most of the cellular PI5P, or is only active under specific conditions. However, after 12 hr treatment with NCT-504, there was a 1.6 fold elevation of PI5P. The fact that this change occurred well after 2 hr of inhibition, suggests that it might not be directly due to an accumulation of the PI5P substrate normally used by PIP4Kγ. Indeed, at 12 hr PI(3,5)P2 levels were also elevated at 2-fold, which was even higher than the fold elevation in PI5P. This raises the possibility that long-term inhibition of PIP4Kγ indirectly results in the activation of PIKfyve. This activation of PIKfyve may account for the increase in PI5P as well as PI(3,5)P2 (*Zolov et al., 2012*; *McCartney et al., 2014*; *Sbrissa et al., 2012*). In addition, at 12 hr PI3P levels increased 1.3 fold, suggesting that VPS34 may be indirectly activated as well.

The elevation of PI3P, PI(3,5)P2 and/or PI5P likely contribute to the elevation in autophagic flux. PI3P has well characterized roles in autophagy, and acts in initiation of phagophore formation (*Shibutani and Yoshimori, 2014*) as well as in later steps of autophagosome maturation (*Carlsson and Simonsen, 2015*), including autophagosome-lysosome fusion (*Ikonomov et al., 2006*) and lysosome reformation (*Rong et al., 2012*; *Yu et al., 2010*). In some conditions, PI5P can induce autophagy independent of PI3P (*Vicinanza et al., 2015*). In contrast, PI(3,5)P2 functions at a late step in autophagy (*Ikonomov et al., 2006*; *de Lartigue et al., 2009*; *Jin et al., 2008*; *Jin et al., 2014*; *Ikonomov et al., 2001*; *Martin et al., 2013*; *Rusten et al., 2007*; *Sano et al., 2016*). These late functions, may contribute to the observed increase in autophagic flux. In addition to these changes the statistically significant decrease in PI4P may also contribute to changes in autophagy. A reduction of PI4P has been postulated to be necessary for lysosome reformation (*Rong et al., 2012*; *Yu et al., 2010*).

The elevation of PI3P, PI(3,5)P2 and PI5P may also have a role in compensating potential mutant huntingtin-dependent changes in PI or masking of selected PI. Several studies have indicated that there are polyglutamine-dependent alterations in PI binding of huntingtin protein (*Burke et al., 2013*; *Kegel et al., 2009a*; *Kegel et al., 2009b*; *Kegel et al., 2005*). Moreover, wild-type huntingtin binds phosphoinositide lipids including PI5P and PI(3,5)P2 (*Kegel et al., 2009b*). Notably, when assessed using unilamellar vesicles, huntingtin with a polyglutamine expansion bound these lipids even more tightly than wild-type hungtingtin, potentially reducing the free total levels of these lipids and impacting their downstream dependent signaling (*Kegel-Gleason, 2013*). Masking of PI lipids could negatively and progressively impact the function of proteins involved in autophagosome cargo recognition and loading, especially those effector proteins dependent on low abundance PI, such as PI5P and PI(3,5)P2. Additional studies need to be carried out to determine the effectors proteins (Alfy and/or others) responsible for the action of PIP4Kγ modulation, the mechanism of action behind the high cellular alteration in PI(3,5)P2 as well as the modulation of other PI levels, and the impact of those changes on mTOR function and autophagy dynamics (*Ikonomov et al., 2006*; *Jin et al., 2014*; *Ikonomov et al., 2001*). It has not escaped our attention that mHtt-dependent effects seem to be triggered by aging, which is known to limit the clearance of misfolded proteins (*Komatsu et al., 2007*), and deregulate phosphatidylinositide lipid signaling (*Igwe and Filla, 1995*).

The data presented in this manuscript demonstrates that pharmacological inhibition, or knockdown of PIP4Kγ produce a similar reduction in huntingtin levels, and a concomitant elevation of PI5P and PI(3,5)P2 and PI3P. These findings open the door to a new disease-modifying approach for this disorder and validate PIP4Kγ as a druggable target. In a recent report, a homozygous mouse knockout displayed no growth or behavioral abnormalities (*Shim et al., 2016*). From the translational point of view, the development of selective PIP4Kγ inhibitors could be extraordinarily useful for other neurodegenerative diseases as well. Alzheimer's disease and Parkinson's disease in particular are also mediated by the accumulation of toxic protein aggregates, whose catabolism by autophagy might rescue stressed neurons. Starvation increases life spans across species (*Speakman and Hambly, 2007*) and there are numerous diseases where upregulation of basal autophagy is beneficial (*Hara et al., 2006*; *Seino et al., 2013*). Further, dose response studies are necessary to fully evaluate the therapeutic potential of PIP4Kγ inhibition.

## Materials and methods

### Synthesis of NCT-504

General Experimental Procedure: Unless otherwise stated, all reactions were carried out under an atmosphere of dry argon or nitrogen in dried glassware. Indicated reaction temperatures refer to those of the reaction bath, while room temperature is noted as ~25°C. All anhydrous solvents, commercially available starting materials, and reagents were purchased from Aldrich Chemical Co. and used as received. Chromatography on silica gel was performed using forced flow (liquid) of the indicated solvent system on Biotage KP-Sil pre-packed cartridges and using the Biotage SP-1 automated chromatography system.

$^1$H spectra were recorded on a Varian Inova 400 MHz spectrometer. Chemical shifts are reported in ppm with the solvent resonance as the internal standard (DMSO-$d_6$ 2.50 ppm, for 1H). Data are reported as follows: chemical shift, multiplicity (s = singlet, d = doublet, t = triplet, q = quartet, br s = broad singlet, m = multiplet), coupling constants, and number of protons.

Analytical purity analysis and retention times (RT) reported here were performed on an Agilent LC/MS (Agilent Technologies, Santa Clara, CA). A Phenomenex Luna C18 column (three micron, 3 × 75 mm) was used at a temperature of 50°C. The solvent gradients are mentioned for each compound and consist of a percentage of acetonitrile (containing 0.025% trifluoroacetic acid) in water (containing 0.05% trifluoroacetic acid). A 4.5 min run time at a flow rate of 1 mL/min was used.

Mass determination was performed using an Agilent 6130 mass spectrometer with electrospray ionization in the positive mode.

Synthetic scheme to prepare NCT-504:

**Scheme 1.** Synthetic scheme to prepare NCT-504.
DOI: https://doi.org/10.7554/eLife.29123.023

Synthetic Procedures:

**B: A** (5-bromo-4-chlorothieno[2,3-d]pyrimidine) was prepared according to WO2012/44993 A1, 2012; Location in patent: Page/Page column 45). A solution of **A** (1.03 g, 4.13 mmol) in THF (15 ml) was treated at 0°C under nitrogen with dropwise addition of isopropylmagnesium chloride (2.48 ml, 4.95 mmol, 2M in THF). The mixture was stirred for 15 min and then a solution of iodine (1.05 g, 4.13 mmol) in THF (10 mL) was added dropwise under nitrogen. The mixture was stirred at 0°C for almost 2 hr, quenched with saturated aqueous $NH_4Cl$ and then EtOAc was added. The mixture was stirred, the organic layer was separated, washed with saturated aqueous $Na_2S_2O_3$, dried with $MgSO_4$, filtered, concentrated to obtain crude 4-chloro-5-iodothieno[2,3-d]pyrimidine (1.17 g, 3.95 mmol, 96% yield). This appeared to be contaminated with a small amount of **B** 4-chlorothieno[2,3-d] pyrimidine (approximately ~5–10% by LC/MS).

1H NMR (400 MHz, DMSO-$d_6$) δ 8.96 (s, 1H), 8.46 (s, 1H).

LC/MS Gradient 4% to 100% Acetonitrile (0.05% TFA) over 3.0 min; RT 3.290 min, ESI (M + 1) +calculated 296.9, found 296.8.

**C:** A microwave vial filled was charged with 4-chloro-5-iodothieno[2,3-d]pyrimidine **B** (0.48 g, 1.62 mmol), (3-(methylsulfonyl)phenyl)boronic acid (0.389 g, 1.94 mmol), Pd(PPh$_3$)$_4$ (0.094 g, 0.081 mmol), sodium carbonate (1.42 ml, 2.83 mmol) followed by dimethoxyethane (8 mL) and water (1 mL). The mixture was heated in the microwave under 'high' settings at 120°C for 20 min in the microwave. The mixture was then cooled; celite was added, and concentrated. The adsorbed material was purified by flash silica gel chromatography with a gradient of 0% to 30% EtOAc in DCM that separated unreacted starting iodide (~20% recovery) from the required product 4-chloro-5-(3-(methylsulfonyl) phenyl)thieno[2,3-d]pyrimidine C (0.19 g, 0.59 mmol, 36% yield).

1H NMR (400 MHz, DMSO-$d_6$) δ 9.02 (d, J = 0.4 Hz, 1H), 8.19 (d, J = 0.4 Hz, 1H), 8.08 (td, J = 1.8, 0.5 Hz, 1H), 8.02 (ddd, J = 7.8, 1.9, 1.1 Hz, 1H), 7.90 (ddd, J = 7.7, 1.7, 1.2 Hz, 1H), 7.77 (td, J = 7.7, 0.5 Hz, 1H), 3.29 (s, 3H). LC/MS Gradient 4% to 100% Acetonitrile (0.05% TFA) over 3.0 min, RT 3.014 min, ESI (M + 1)+ calculated 325.0, found 324.9.

**NCT-504** 4-Chloro-5-(3-(methylsulfonyl)phenyl)thieno[2,3-d]pyrimidine C (0.18 g, 0.55 mmol) with DME (10 mL) was treated with 1-methyl-1H-tetrazole-5-thiol (0.084 g, 0.720 mmol) and Hunig's Base (0.194 mL, 1.11 mmol), and heated at 120°C for 30 min in a sealed tube. The mixture was cooled, concentrated, re-dissolved in minimal DCM and the purified by silica gel column chromatography (5% to 60% EtOAc/DCM) to provide NCT-504 4-((1-methyl-1H-tetrazol-5-yl)thio)−5-(3-(methylsulfonyl)phenyl)thieno[2,3-d]pyrimidine (190 mg, 0.470 mmol, 85% yield).

1H NMR (400 MHz, DMSO-$d_6$) δ 8.76 (d, J = 0.4 Hz, 1H), 8.23 (td, J = 1.8, 0.5 Hz, 1H), 8.15 (d, J = 0.4 Hz, 1H), 8.10 (ddd, J = 7.8, 1.9, 1.1 Hz, 1H), 8.02 (ddd, J = 7.7, 1.7, 1.1 Hz, 1H), 7.86 (td, J = 7.8, 0.6 Hz, 1H), 3.96 (s, 3H), 3.34 (s, 3H).

LC/MS Gradient 4% to 100% Acetonitrile (0.05% TFA) over 3.0 min, RT 3.024 min, ESI (M + 1)+ calculated 405.0, found 405.0.

## Enzyme preparation and biochemical assays

Protein from human *PIP4K2A* (UniGene 138363), *PIP4K2B* (Unigene 269308) and *PIP4K2C* (UniGene 6280511) was expressed in pGEX6P (GE Healthcare) and purified from *E. coli* BL21(DE3). GST fusion proteins from cell lysates were bound to glutathione sepharose beads (GE Healthcare) and cleaved in situ with 50U of PreScission protease (GE Healthcare) for 4 hr at 4°C.

Lipid kinase assays were performed essentially as described previously (*Wang et al., 2010*; *Clarke et al., 2001*). In brief, dried substrate lipid (6 μM PI5P final reaction concentration) was resuspended in kinase buffer (50 mM Tris pH 7.4, 10 mM $MgCl_2$, 80 mM KCl, and 2 mM EGTA) and micelles were formed by sonication for 2 min. Recombinant lipid kinase, preincubated with inhibitor for 10 min on ice (where required), was added to the micelles and the reaction started by the addition of 10 μCi [$^{32}$P]ATP (200 μl final volume), and incubated at 30°C for 10–60 min (dependent on isoform). Lipids were extracted using an acidic phase-separation (*Bligh and Dyer, 1959*) and separated by one-dimensional thin layer chromatography (2.8:4:1:0.6 chloroform:methanol:water:ammonia). Radiolabelled PI(4,5)P2 product was detected by autoradiography, extracted from the plate and Cerenkov radiation was counted in the presence of Ultima Gold XR scintillant (Packard) on a LS6500 scintillation counter (Beckman Coulter). Specific enzyme activities, under these assay conditions, were calculated as nmoles of PI5P converted into PI(4,5)P2 per minute per mg of purified recombinant enzyme.

Intrinsic ATPase activities of the enzymes were determined using the Transcreener $ADP^2$ fluorescence polarization assay (BellBrook Labs). PIP4Kγ (1 μM, [*Clarke and Irvine, 2013*]) was pre-incubated (10 min on ice) at range of inhibitor concentrations and assayed with ATP substrate (100 μM ATP, 60 min incubation at 22°C) in the absence of lipid substrate. Polarization units (mP) were read using a PHERAstar Plus microplate reader (BMG Labtech). Experimental values were interpolated from an ADP/ATP utilization standard curve and plotted using nonlinear regression analysis with Prism 5 (GraphPad).

## Measurement of phosphorylated phosphoinositide (PI) levels by HPLC

PI measurements were performed as previously described (*Zolov et al., 2012*). Briefly, mouse primary fibroblasts were generated from P1 pups (129P2/OlaHsd × C57BL/6) and were cultured in DMEM supplemented with 15% FBS and 1X Pen-Strep-Glutamine and human patient fibroblast were cultured in MEM supplemented with 15% FBS, 1x Pen-Strep and 1x Glutamax in 100 mm dishes to 60–70% confluence. MEF cells and patient fibroblasts were tested using MycoFluor Mycoplasma Detection Kit (Thermo Scientific Fisher) and were negative for mycoplasma contamination. Cells were washed with PBS and incubated with inositol labeling medium (containing custom-made inositol-free DMEM (11964092; Life Technologies), 10 μCi/mL of myo-$^3$H-inositol (GE Healthcare), 10% dialyzed FBS (26400; Life Technologies), 20 mM Hepes, pH 7.2–7.4, 5 μg/mL transferrin (0030124SA; Invitrogen), and 5 μg/mL insulin (12585–014; Invitrogen) for 48 hr. For experiments with NCT-504 treatments, cells were treated with indicated concentrations of NCT-504 or DMSO for indicated

duration before the end of the labeling. Extraction and HPLC measurements were performed as described (*Zolov et al., 2012*).

## Silencing of PIP4Kγ

Primary mouse embryonic fibroblast cells generated from P1 pups (129P2/OlaHsd × C57BL/6) were infected with MISSION shRNA lentiviral plasmid pLKO.1-puro with shRNA target sequence C TCCAAGATCAAGGTCAACAA (TRCN0000024702; Sigma) containing 237–257 nucleotides of mouse PIP4Kγ cDNA; MISSION nontarget shRNA lentiviral control vector SHC002 (Sigma) was used as control. Transduction-ready viral particles were produced by the Vector Core (University of Michigan, Ann Arbor, MI) with a concentration of $10^7$ transduction units per ml. Mouse primary fibroblast grown on two 35 mm dishes were treated at an MOI of 5. After overnight incubation, cells were treated with 2 µg/ml puromycin. After two days of infection, cells from two 35 mm dishes were transferred to a 100 mm dish and maintained in puromycin containing media for another three days. Cells were either analyzed by western blot or incubated with inositol labeling medium for 48 hr for PI measurements. Immunoblots were performed with antibodies against PIP4K2C (17077–1-AP RRID: AB_2715526, ProteinTech; 1:5000) and GADPH (AM4300 RRID: AB_437392, Thermo Scientific Fisher; 1:50000).

## LC3 measurements in HEK cells

HEK 293T cells grown on 35 mm Dishes till 60–70% confluency were either untreated or treated with DMSO or NCT-504 with or without 100 nM Bafilomycin for two hours. Cells were lysed and immunoblotted with antibodies against LC3A/B (12741 RRID:AB_2617131; Cell Signaling) and α-tubulin (A-11126 A11126 RRID:AB_221538; Life Technologies). Blots were analyzed using Adobe Photoshop. HEK293T cells were purchased from ATCC (RRID:CVCL_0063) and were certified authentic and mycoplasma free.

## Htt HTRF assay

### Antibodies

The monoclonal antibodies used in the HTRF assay were 2B7 (gift from collaborator) which binds to the first 17 amino acids of normal and mutant Htt, and MAB2166 (EMD Milipore #MAB2166), which binds to an Htt epitope (amino acid 181 to 810), and recognizes both normal and mtHtt. The antibody 2B7 was conjugated to Tb as a donor, and 2166 was conjugated to d2 as an acceptor (both were custom labeled by Cisbio). The labeled antibody pairs were diluted in the 1X HTRF assay buffer: 50 mM $NaH_2PO_4$, 400 mM NaF, 0.1%BSA, 0.05% Tween 20. The HTRF assay were performed at 1536-well plate. For the experiments, cells were seeding (6 µL/well) 24 hr in advanced and culture at 37°C 5%$CO_2$ followed by compound addition (23 nL). After incubating with compounds for 24 hr, cells were lysed by adding 2 µL of 4x lysis buffer (Cisbio Lysis buffer #2), incubated at room temperature for 2 hr then add labeled antibodies. The labeled antibody pairs were diluted in the 1X HTRF assay buffer: 50 mM $NaH_2PO$, 400 mM NaF, 0.1%BSA, 0.05% Tween 20. The final reaction is 8 µL/well. The signal ratio between 665 nm and 615 nm have been calculated as the raw HTRF ratio. The cell viability was measured by using CellTiter-Glo Luminescent Cell Viability Assay (Promega). The 293A cells were purchased from ThermoFisher Scientific, Cat#R70507, Lot # 1657360. They tested negative for mycoplasma. They were not sent out for STR since it was first use right after purchase from company.

## Htt exon1 aggregation assay

GFP-Htt-exon1-Q23 (Plasmid #40261)1 and GFP-Htt-exon1-Q74 (Plasmid #40262)1 were purchased from Addgene (Cambridge, MA) (*Narain et al., 1999*). Immortalized Atg7+/+ and Atg7-/- MEF cells were generously provided by Dr. Masaaki Komastu (School of medicine, Niigata University) (*Komatsu et al., 2005*). Atg7+/+ and Atg7-/- MEF cells were tested using MycoFluor Mycoplasma Detection Kit (Thermo Scientific Fisher) and are negative for mycoplasma contamination. They were validated for the presence and absence, respectively, of Atg7 by the western blot shown in *Figure 3C*. These cell lines are not included in the list of commonly misidentified cell lines maintained by International Cell Line Authentication Committee were not authenticated further. HEK293T, Atg7+/+ and Atg7-/- cells grown on coverslips were transfected with either GFP-Htt-

exon1-Q23 or GFP-Htt-exon1-Q74 using Lipofectamine 2000 (Invitrogen). After 2 hr of transfection, cells were incubated with DMSO or 2 µM of NCT-504 for 48 hr and fixed. Transfected cells with mHtt aggregates were quantified (*Narain et al., 1999*; *Komatsu et al., 2005*).

## HTT quantification in fibroblasts and StHdh cells

Immortalized StHdhQ111 (Coriell-CH00095, RRID:CVCL_M591) cells (*Trettel et al., 2000*), immortalized wild type (Coriell-GM02153) and HDQ45 (Coriell-GM03868, RRID:CVCL_1H73) HD fibroblasts ((using SV40 large T antigen) (*Lu et al., 2013*) and non-immortalized HDQ68 (Coriell-GM21757, RRID:CVCL_1J85) were grown in 15%FBS DMEM with GlutaMax (Life Technologies). For drug treatments, cells were plated overnight until they reached 70% confluence in 12-well plates, drug was added at the desired concentration for 48 hr. For siRNA treatment cells were nucleofected using Amaxa at a final concentration of 30 nM and grown in 6-well plates for 72 hr. StHdh cells were grown in DMEM (Life Technologies) 10% FBS and drug treatment was carried out as described above. Cell identity was confirmed using STR profiling (GenePrint 10 System from Promega Corp.) and tested mycoplasma negative (Hoechst staining).

Cells were collected using trypsin, homogenized in RIPA buffer, sonicated and incubated in ice for 30 min. Supernatant was collected after a 10 min centrifugation and protein concentration was measured. For western blot analysis 15 µg of each protein sample was loaded in a 4–12% Bis-tris gel, transferred into a nitrocellulose membrane, blocked with 5% milk and incubated overnight with anti-Huntingtin antibody MAB5492 (Millipore) for fibroblasts or MAB2166 (Millipore) for StHdhQ111 cells.

## Ethical treatment of animals

All vertebrate animal work was approved by the Institutional Animal Use and Care Committee at the University of Michigan (PRO00007096). Experiments were carefully planned to minimize the number of animals needed. Pregnant female wild-type, non-transgenic Long Evans rats (*Rattus norvegicus*) were housed singly in chambers equipped with environmental enrichment. They were fed ad libitum a full diet (30% protein, 13% fat, 57% carbohydrate; full information available at www.labdiet.com), and cared for by the Unit for Laboratory Animal Medicine (ULAM) at the University of Michigan. Veterinary specialists and technicians in ULAM are trained and approved in the care and long-term maintenance of rodent colonies, in accordance with the NIH-supported Guide for the Care and Use of Laboratory Animals. All rats were kept in routine housing for as little time as possible prior to euthanasia and dissection, minimizing any pain and/or discomfort. Pregnant dams were euthanized by CO2 inhalation at gestation day 20. For each animal, euthanasia was confirmed by bilateral pneumothorax. Euthanasia was fully consistent with the recommendations of the Guidelines on Euthanasia of the American Veterinary Medical Association and the University of Michigan Methods of Euthanasia by Species Guidelines. Following euthanasia, the fetuses were removed in a sterile manner from the uterus and decapitated. Primary cells from these fetuses were dissected and cultured immediately afterwards.

## Rodent primary neuron isolation and culturing

Primary mixed cortical neurons were dissected from these embryos as described previously (*Saudou et al., 1998*), and plated in a poly-l-lysine/laminin coated 96 well plate at a density of $5 \times 10^5$ cells/ml. On day four in vitro, cells were transfected with Dendra2-LC3 with or without GFP-Beclin using Lipofectamine 2000 (Invitrogen). Thirty minutes post-transfection, neurons were treated with NCT-504 or DMSO. Optical pulse labeling experiments were performed as previously described (*Tsvetkov et al., 2013*; *Gupta et al., 2017*; *Barmada et al., 2014*). Briefly, Dendra2-LC3 was photoconverted 24 hr post-transfection by illuminating each imaging field with a 250 ms pulse of 405 nm light. Following photoconversion, neurons were longitudinally imaged using a custom-built automated fluorescence microscopy platform (*Arrasate et al., 2004*; *Barmada et al., 2014*; *Barmada et al., 2015*). A Nikon Eclipse Ti inverted microscope equipped with a high-NA 20X objective lens, a PerfectFocus3 system, and an Andor iXon3 897 EMCCD camera were used for image acquisition. GFP and TRITC images were taken immediately after photoconversion and four more times within the following 48 hr. Single-cell TRITC intensity values were fitted to a first-order exponential decay curve, generating a half-life value for each individual neuron. Neuronal survival analysis

was assessed using original software written in Python. Only cells that lived the duration of imaging were included in the Dendra2-LC3 half-life analysis. Half-life was determined by fitting the TRITC intensity values at each time point to a first-order exponential function using scripts written in R. Comparisons between groups to determine statistical significance were accomplished using one-way ANOVA with Dunnet's post hoc test and the Kruskal-Wallis test.

### *Drosophila* experiments

Two different *Drosophila* HTT-expressing strains were used for this study, and N-terminal model expressing the first 336 amino acids of human HTT with a 128Q expansion (*Branco et al., 2008*) and a full length model expressing human HTT with a Q200 expansion (*El-Daher et al., 2015*). For retinal expression, we used the GMR-GAL4 driver at 25C and for panneuronal expression, we used the elav-GAL4 driver. These two drivers as well as the siRNAs targeting dPIP4K were obtained from the Bloomington *Drosophila* stock center. The dPIP4K-29 loss of function allele and the K271D kinase dead (PIP4K-DN) allele were previously described and kindly provided by Dr. Padinjat Raghu (*Gupta et al., 2013*).

For the retinal degeneration assay, animals were fixed with 4% formaldehyde in PBS. Heads were dehydrated in increasing concentrations of ethanol and embedded in paraffin. Ten μm serial sections were obtained and re-hydrated to PBS. Sections were stained with hematoxylin (SIGMA). Images were captured using an AxioCam MRc camera (ZEISS) attached to a MICROPHOT-FXA microscope (Nikon).

Motor performance of animals was assessed as a function of age. For the N-terminal model 15 age-matched virgin females per replica were used. Animals are taped to the bottom of a plastic vial and the number of animals reaching a height of 9 cm in 15 s is assessed using infrared sensors. Ten trials are carried out for each day represented. The plotted data corresponds to the average percentage of animals reaching 9 cm. Data was analyzed by ANOVA followed by Dunnet's post hoc test. For the FL-HTTQ200 a similar procedure was used, the animals were video recorded and data was processed using a custom designed analysis software (*Source code 1*), which allowed for calculating speed.

## Acknowledgements

Immortalized MEF wild-type and MEF Atg7 knock-out cells were a gift from Dr. Masaki Komatsu (Niigata University, Japan). None of the cell lines used in this study were included in the list of commonly misidentified cell lines maintained by International Cell Line Authentication Committee. This work was supported in part by National Institutes of Health (NIH) grants R01-NS064015 and R01-NS099340 to LSW, R01-NS097542 and P30-AG053760 to SJB, and the Protein Folding Diseases Fast Forward Initiative, University of Michigan to LSW and SJB. SSPG was supported in part by AHA Postdoctoral Fellowship, 14POST20480137. IA was supported by R21NS096395 grant from the NIH and by the Darrell K Royal Research Fund for Alzheimer's Disease. JB was supported by grants from the CHDI and the Robert A. and Renée E. Belfer Family Foundation.

## Additional information

### Funding

| Funder | Grant reference number | Author |
| --- | --- | --- |
| Foundation for the National Institutes of Health | R21NS096395 | Ismael Al-Ramahi |
| Darrel K Royal Research Fund for Alzheimers Disease | | Ismael Al-Ramahi |
| American Heart Association | Post Doctoral Fellowship (14POST20480137) | Sai Srinivas Panapakkam Giridharan |
| National Institutes of Health | P30-AG053760 | Sami Barmada |
| National Institutes of Health | R01-NS097542 | Sami Barmada |

| University of Michigan | Protein Folding Diseases FastForward Initiative | Lois S Weisman Sami Barmada |
| --- | --- | --- |
| National Institutes of Health | R01-NS064015 | Lois S Weisman |
| National Institutes of Health | R01-NS099340 | Lois S Weisman |
| CHDI Foundation | | Juan Botas |
| Robert A. and Renee E. Belfer Family Foundation | | Juan Botas |

The funders had no role in study design, data collection and interpretation, or the decision to submit the work for publication.

## Author contributions

Ismael Al-Ramahi, Conceptualization, Data curation, Formal analysis, Supervision, Visualization, Methodology, Writing—original draft, Writing—review and editing; Sai Srinivas Panapakkam Giridharan, Conceptualization, Data curation, Formal analysis, Validation, Investigation, Visualization, Methodology, Writing—original draft, Writing—review and editing; Yu-Chi Chen, Amanda K Wagner Gee, Investigation, Visualization, Methodology; Samarjit Patnaik, Data curation, Formal analysis, Investigation, Methodology, Writing—original draft, Project administration, Writing—review and editing; Nathaniel Safren, Investigation, Methodology, Writing—original draft; Junya Hasegawa, Conceptualization, Data curation, Formal analysis, Investigation, Visualization, Methodology, Writing—review and editing; Maria de Haro, Data curation, Formal analysis, Validation, Investigation, Visualization, Methodology; Steven A Titus, Validation, Investigation, Visualization, Methodology; Hyunkyung Jeong, Formal analysis, Investigation, Visualization, Methodology; Jonathan Clarke, Investigation, Methodology, Writing—review and editing; Dimitri Krainc, Marc Ferrer, Conceptualization, Resources, Formal analysis, Supervision, Funding acquisition, Methodology; Wei Zheng, Conceptualization, Resources, Supervision, Funding acquisition, Methodology, Writing—review and editing; Robin F Irvine, Conceptualization, Resources, Methodology, Writing—original draft; Sami Barmada, Resources, Supervision, Funding acquisition, Investigation, Methodology, Writing—review and editing; Noel Southall, Data curation, Formal analysis, Writing—original draft, Writing—review and editing; Lois S Weisman, Juan Jose Marugan, Conceptualization, Resources, Data curation, Formal analysis, Supervision, Funding acquisition, Validation, Investigation, Visualization, Methodology, Writing—original draft, Project administration, Writing—review and editing; Juan Botas, Conceptualization, Resources, Data curation, Formal analysis, Supervision, Funding acquisition, Validation, Investigation, Visualization, Methodology, Writing—original draft, Project administration

## Author ORCIDs

Samarjit Patnaik http://orcid.org/0000-0002-4265-7620
Junya Hasegawa http://orcid.org/0000-0002-7041-890X
Jonathan Clarke http://orcid.org/0000-0002-4079-5333
Noel Southall http://orcid.org/0000-0003-4500-880X
Juan Jose Marugan http://orcid.org/0000-0002-3951-7061

## Ethics

Animal experimentation: All vertebrate animal work was approved by the Institutional Animal Use & Care Committee at the University of Michigan (PRO00007096).

## Decision letter and Author response

Decision letter https://doi.org/10.7554/eLife.29123.027
Author response https://doi.org/10.7554/eLife.29123.028

## Additional files

### Supplementary files

• Source code 1. Custom Software for statistical analysis.

DOI: https://doi.org/10.7554/eLife.29123.024

• Transparent reporting form
DOI: https://doi.org/10.7554/eLife.29123.025

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
