## [Decision Letter]

[Editors’ note: a previous version of this study was rejected after peer review, but the authors submitted for reconsideration. The first decision letter after peer review is shown below.]

Thank you for submitting your work entitled "Inhibition of PIP4Kγ ameliorates the pathological effects of mutant huntingtin protein" for consideration by *eLife*. Your article has been reviewed by three peer reviewers, one of whom, Harry T Orr (Reviewer #1), is a member of our Board of Reviewing Editors, and the evaluation has been overseen by a Senior Editor. The following individuals involved in review of your submission have agreed to reveal their identity: Leslie Thompson (Reviewer #3).

Our decision has been reached after consultation between the reviewers. Based on these discussions and the individual reviews below, we regret to inform you that your work will not be considered further for publication in *eLife*.

Note that while reviews found considerable merit in your study, there are major limitations that need to be addressed in order for *eLife* to consider your paper in the future. First, it is important that these studies be performed using a full-length Htt fly model. Second, it is also critical that your analysis include a dosing regime of the PIP4K compound. Should you choose to perform these studies and submit the new manuscript in the future, we will attempt to get some of the same reviewers.

Reviewer #1:

In this study the investigators present a rather extensive body of work (and concise) in support of the concept that inhibition of the lipid kinase PIP4Kγ reduces levels of huntingtin and mitigates phenotypic effects of mutant Htt in a fly model, presumably via induction of autophagy. Building on their previous work, a novel PIP4K inhibitor and its ability to enhance autophagy in cells and reduce Htt levels in neurons are presented. Importantly they go on to show that in a fly model of mutant Htt pathology reduction of PIP4Kγ mitigates two disease-related phenotypes; retinal degeneration and a motor performance deficit. All in all the data are solid and present a strong basis for exploring further the therapeutic potential of targeting PIPK4γ.

Reviewer #2:

This manuscript is describing the inhibition of PIP4Kγ as a potential therapeutic target in Huntington's disease by the triggering of autophagy to clear huntingtin protein, in what they refer to as a druggable target.

This small molecule lead, NCT-504, is important because it appears to trigger the degradation of huntingtin, and thus falls into the hypotheses of therapies for HD designed to _specifically_ reduce or eliminate huntingtin protein in Huntington's disease, which include AAV siRNAs, and anti-sense oligonucleotides, which are either pending or are in clinical trials currently. The concept behind this study was driven by mouse exon1 model work that implies that increasing the rate of autophagy may be beneficial for neurodegeneration in HD.

The manuscript has some solid data, but is confusing to follow in that they go back and forth from small fragment over expression models, in which 3% of the mutant huntingtin ORF with synthetic allele is over expressed at massive levels to induce the formation of aggregates, to data from human HD fibroblasts, which is arguably the most exciting data in this work. The only assay of pathological effects comes from an outdated fly fragment model.

The overall picture seems exciting but the details in the data are not as clear as the writing would suggest. In many of the figures and assays, they only look at mutant HD cells, and not fibroblasts from unaffected individuals, and most of the readouts are only looking at one protein, while the proposed mechanism is to switch on autophagy in general. Autophagy, and its sub-categories of mitophagy, micro-autophagy, macro-autophagy, pexophagy are all dynamic system in a high rate of constant flux in the cell. If this is truly a drug lead for Huntington's disease, or any disease, they need to show some selectivity in the effected state and in the target. It seems that even in their loading controls, there is a massive global effect of shutting down PIP4Kγ either by SiRNA or global inhibitor. In other words, autophagy in all its forms is in dynamic flux for a reason, and just pushing it in one direction may be as problematic and inhibiting it.

I have an conceptual concern that this data suggest that 97% of the huntingtin ORF is irrelevant for regulating the turnover of this protein by autophagic mechanisms, and it appears from this data that the molar stoichiometry is irrelevant, as whether they look at endogenous huntingtin proteins, or massively over-expressed small molecule fragments at orders of magnitude above physiological molarity, the compound works just as well to clear 'huntingtin', and likely everything else.

My major concern is that I believe they have proven that PIP4K γ inhibition triggers a massive level of autophagy, but they have no data here that show any selectivity or health effects on the fibroblasts. Is this "curing dandruff by decapitation"? They reference a manuscript on a PIP4Kγ knockout mouse stating the investigators observed "no growth or behavioral abnormalities", except the manuscript actually describes a massive global hyperactivation of the immune systems in those knockouts, and the true validation of a drug target would be a conditional knockout. For what it's worth, Broad's Exac exomic sequencing database shows no homozygous null alleles of PIP4Kγ yet in humans. Thus, I have concerns that PIP4K γ inhibition could really be a drug target.

They do not state exactly how their HD fibroblasts are immortalized, but is seems that there is no difference in effect between HEK293, STHdHQ111, HD primary cells, or HD immortalized cells. This is rather surprising since the only known inducer of huntingtin expression is P53, and some of those lines have P53, and others do not. This has been a concern that has been ignored in most HD cell lines studies, despite over 80 manuscripts published consistently defining disrupted P53 biology in HD, from mouse to humans. So whether they have p53 or not, whether small fragments of full protein, everything gets cleared, in all lines.

Major concerns

Figure 1

From the high throughput screen, they found NCT-504 and by the structure guessed that it was a kinase inhibitor, and tested this by kinomic screening, and used a cutoff of >65% inhibition to claim specificity. This is a very high cutoff, and the entire kinome screening data should be included a supplemental, not just 3/442 kinases. From my experience with pharma, even 20% inhibition is cause for concern on these screens. The IC50 is about 16μM for NCT-504. This is very high, and by their own data, the IC50 exceeds the toxicity threshold concentration in neurons.

The concentrations of NCT-504 used for effects vary hugely across the figures. In Figure 1, and from text, IC50 is quoted at 15.8μM. In Figure 1—figure supplement 1, Figure 1—figure supplement 2μM is used to reduce aggregation by 50% effect, so whatever reason they see effect, it seems to be unlikely through PIP4Kγ inhibition.

Figure 2: concentration of NCT-504 is now 10μM, less than the IC50, and only one dose tested.

The loading controls seem to switch between α-tubulin, actin and GAPDH. For these studies looking at autophagic flux, as in Figure 4 they need better controls, not an insoluble polymer of actin. I'd like to see GAPDH and at least two inner mitochondrial proteins to show me the cells are not just completely autophagic or apoptotic. Images are essential.

Again, for Figure, 4, now 2μM of NCT-504 is used again in Q68 cells, but then 10μM in Q45 cells. This is plus or minus 500%.

Figure 4: they need to show the whole blot in a supplemental figure. I see a severe hit on actin levels in Figure 4 with PIP4K2C_si, and in Figure 4 with 10μM NCT-504, which concerns me that these cells might be heavily auto-degrading or just dead.

In Figure 4—figure supplement 4, they show a very unconventional assay for neuronal toxicity, in that they are no showing any images of neurons or other measure of health, or even life, but their own data shows a pretty severe "toxicity" at 10μM NCT-504, the exact concentration used in HD fibroblasts, and still below their calculated IC50.

Figure 4—figure supplement 4, that western blot is not appropriate data to conclude anything.

Figure 5. They now switched back to a severe allele Q128 (extremely rare) and only small fragment 1-231. Why did they not use full huntingtin?

Reviewer #3:

This is a very well written manuscript that describes strong data showing that PIP4K γ inhibition, either pharmacologically or through KD, causes reduction of levels of soluble HTT. In cases where aggregation or formation of HMW species are observed, PIP4K reduction also reduces these aggregated species as well. The authors used several different cell lines ranging from immortalized lines to primary neurons or human fibroblasts (N2a, stHDHq111, human fibroblasts, primary neurons). Experiments are elegant, well described and well controlled, with helpful supplemental data, such as the KD efficiency of the RNAis and kinetics of the phosphatidylinositide changes with treatment. There are a few minor concerns:

1) How was NCT-504 selected for these studies from published the high-throughput screen? Was this best hit? Because of the complexity of these pathways, it would be helpful to have a schematic early on to show the various components of the pathway, how each of the PIP derivatives are generated and where this kinase fits into general scheme.

2) Figure 4 - Why not show more complete set of westerns as in C and D? Need quantification for Figure 4.

3) Do the fibroblasts have HMW species? This is not mentioned and if they are present, why they are not included should be addressed.

4) A potentially important experiment would be using a pulse chase or other such assay to measure whether this truly increases degradation of the protein.

5) Please describe length of HTT fragment in the 128Q HTT transgene in flies.

6) A citation should be included for the Neuron paper from the Davidson lab when discussing Rhes and Rheb.

7) It is a stretch to say that compound modulates HTT proteolysis as there is a single Western blot with one panel.

[Editors’ note: what now follows is the decision letter after the authors submitted for further consideration.]

Thank you for submitting your article "Inhibition of PIP4Kγ ameliorates the pathological effects of mutant huntingtin protein" for consideration by *eLife*. Your article has been reviewed by three peer reviewers, one of whom is a member of our Board of Reviewing Editors, and the evaluation has been overseen by a Senior Editor. The reviewers have opted to remain anonymous.

The reviewers have discussed the reviews with one another and the Reviewing Editor has drafted this decision to help you prepare a revised submission.

Summary

In this manuscript the ability of a small molecule to inhibit PIP4Kγ and impact HD-like phenotypes in cellular and animal models expressing mutant Htt (full-length and fragment) are presented. While one reviewer is concerned with the extent to which these findings might apply to a mammalian model of HD, it was noted by other reviewers that their use of primary cortical neurons from mice provides strong evidence of the biological relevance of this compound and pathway beyond its heavy use of fly HD models.

Essential revisions

It is critical that the authors pay close attention to and respond accordingly to reviewers 1 and 2 concerns that the strength of this manuscript is decreased considerably by the style/manner in which it is written. Extensive editing to the text and figures are needed so that the manuscript flows smoother and rationale for the studies are clearer.

Reviewer #1:

In this new submission of a previously submitted manuscript the ability of a small molecule to inhibit PIP4Kγ and impact HD-like phenotypes in cellular and animal models expressing mutant Htt (full-length and fragment) is presented. Overall while this work addresses a very interesting issue and additional data using a FL-Htt fly model have been added, it remains difficult to read/follow in several key places.

Specific points:

Overall the manuscript remains hard to read, needing English editing in many places.

In several places statements are not referenced when they should be. For example, second paragraph of Results – "Similar discrepancies in potency […]" needs to be referenced.

Clarity of Figure 3 would be improved by including concentrations of compounds used within body of the figure and not just in the legend.

Reviewer #2:

This is a revised manuscript and the authors were very responsive to the reviewers.

The work described identifies PIP4Kγ as a therapeutic target for HD and shows that pharmacological inhibition of the enzyme promotes mutant HTT degradation and autophagy. They provide in vivo evidence that lower the levels of PIP4Kγ is beneficial in a fly model of HD-retinal assays and motor performance. This is a promising direction to pursue as a possible therapy for HD and is in line with current treatments directed at lowering mHTT protein levels. Generally the quality of the data is strong. Minor points are that there are a number of formatting typos. Also the figures are not very consistent. Formatting is varied-different fonts, different graphing and repetitive information in legend and graphs.

Reviewer #3:

This manuscript describes the effect of inhibition of the PIP4Kγ in models of Huntington's disease. There is extensive description of the effect of the drug on lipid metabolism and autophagy. There are also data from cell models including HD fibroblasts. The reduction of Htt levels is of interest. However effects on Htt aggregates are difficult to interpret since aggregates may be toxic or protective. There is also an interesting effect on an HD *Drosophila* model. It would be beneficial if the investigators had some more substantial data on models more directly relevant to HD pathogenesis. For instance instead of using HD fibroblasts it would be beneficial to use HD iPS cell models. In addition, there are no data from HD mouse models. The processes of autophagy are cell type specific, and may be different in *Drosophila* and mammalian systems, so in vivo data from mice rather than flies would be advantageous.

---

## [Author Response]

[Editors’ note: the author responses to the first round of peer review follow.]

[…] Note that while reviews found considerable merit in your study, there are major limitations that need to be addressed in order for eLife to consider your paper in the future. First, it is important that these studies be performed using a full-length Htt fly model.

We have included now results evaluating the effect of PIP4K loss of function in a *Drosophila* model expressing full length Htt (new panels in Figure 5). As shown decreasing *dPIP4K(CG17471)* by either a heterozygous mutant (PIP4K-dCG17471^LOF^) or by expression of a dominant negative allele (PIP4K-dCG17471^DN^) leads to a suppression of the HTT-FLQ200-induced motor impairments. We used two different parameters to measure motor performance: 1- a climbing test and 2- average climbing speed.

Second, it is also critical that your analysis include a dosing regime of the PIP4K compound.

In our original manuscript we included several concentration-response experiments showing the effect of our PIP4Kγ inhibitor at different dose ranges that now we have reinforced with additional experiments:

· The kinase inhibitor IC50 of NCT-504 was reported using two different methods: binding assay (DiscoverX KINOMEscan, Figure 1) and lipid phosphorylation using full length isolated PIP4Kγ (Figure 1).

· We also reported the full concentration-response curve of NCT-504 inhibiting mHtt cellular accumulation in stably transfected PC12 cells with a Q103 mHtt fragment, fused to GFP (Figure 1).

· In addition, we have also included a dose response for autophagosome formation and autophagy flux using a stable reporter HEK293T cell line transfected with LC3-GFP-mCherry.

· The same cell line HEK293T was used with 2-different concentrations of NCT-504 in the presence and absence of bafilomycin A1 to evaluate LC3 levels.

· We have also evaluated the effect of NCT-504 at different doses on autophagy flux in rat primary neurons carrying LC3-Dendra2, a photoswitchable protein able to report LC3 turnover in individual neurons. With this technique previous authors have shown that the proteostasis of Htt-poly-Q varies among neurons and predicts neurodegeneration.

· We have also performed titration curves for NCT-504 in the Q45 patient fibroblasts. As shown in Figure 4—figure supplement 1, NCT-504 is not toxic until we reach the 10μM concentration. The dose response experiments on mHTT protein levels revealed that there was a statistically significant decrease of mHTT levels starting at 5μM (new representative western blot and quantification now included in Figure 4).

· The dose-dependent effect of NCT-504 in phosphatidylinositols levels has been also included using human fibroblast.

Reviewer #2:This manuscript is describing the inhibition of PIP4Kγ as a potential therapeutic target in Huntington's disease by the triggering of autophagy to clear huntingtin protein, in what they refer to as a druggable target.This small molecule lead, NCT-504, is important because it appears to trigger the degradation of huntingtin, and thus falls into the hypotheses of therapies for HD designed to _specifically_ reduce or eliminate huntingtin protein in Huntington's disease, which include AAV siRNAs, and anti-sense oligonucleotides, which are either pending or are in clinical trials currently. The concept behind this study was driven by mouse exon1 model work that implies that increasing the rate of autophagy may be beneficial for neurodegeneration in HD.

We have included additional references to provide more context around this line of research.

· Sarkar S1, Perlstein EO, Imarisio S, Pineau S, Cordenier A, Maglathlin RL, Webster JA, Lewis TA, O'Kane CJ, Schreiber SL, Rubinsztein DC., Small molecules enhance autophagy and reduce toxicity in Huntington's disease models, Nat Chem Biol. 2007 Jun;3(6):331-8. Epub 2007 May 7.

· Lin F1, Qin ZH1, Degradation of misfolded proteins by autophagy: is it a strategy for Huntington's disease treatment?, J Huntingtons Dis. 2013;2(2):149-57. doi: 10.3233/JHD-130052.

The manuscript has some solid data, but is confusing to follow in that they go back and forth from small fragment over expression models, in which 3% of the mutant huntingtin ORF with synthetic allele is over expressed at massive levels to induce the formation of aggregates, to data from human HD fibroblasts, which is arguably the most exciting data in this work. The only assay of pathological effects comes from an outdated fly fragment model.

For historical and practical reasons, our primary screen was carried out in a cell-based aggregation readout screen. This model allowed us to quickly screen libraries for compounds that could modulate the levels of NT-mHtt fragment. A similar approach would have been impractical in other cell models including the human HD fibroblasts. Once we identified NCT-504 as a potential target in the primary screen, we proceeded to the validation of the compound as well as its putative target (PIP4Kγ) in more physiologically relevant models, because we completely agree with the reviewer that these models are better suited for understanding mHtt biology. Given the consistency we observe between the fragment models and the full length human patient fibroblasts or mouse STHdh^Q111^ we think that the screening strategy is validated. Although currently not a feasible option, in future studies we are aiming at performing these protein levels screens in the HD patient fibroblasts.

With regard to the *Drosophila* model, we have followed the reviewers’ advice by evaluating the effect of PIP4K loss of function in a *Drosophila* model expressing full length Htt with 200 glutamines (new panels in Figure 5). As shown decreasing *dPIP4K(CG17471)* by either a heterozygous mutant (PIP4K-dCG17471^LOF^) or by expression of a dominant negative allele (PIP4K-dCG17471^DN^) leads to a suppression of the Htt-FLQ200-induced motor impairments. We used two different parameters to measure motor performance: 1- a climbing test and 2- average climbing speed.

Our previous submission presented data using an N-terminal exon 1 fragment Htt-poly-Q model. We now include data using full length Htt-poly-Q, obtaining similar results and indicating that silencing of the gene or elimination of a single allele ameliorates the neurodegeneration and loss in motor skills. There are numerous published studies by us and others using *Drosophila* models (PMID: 18184562, 18184562) and mouse models (PMID: 24052178) of Huntington disease expressing full length huntingtin poly-Q or just the N-terminal exon 1 fragment. Comparing the phenotype of the disease in both models, exon 1 Htt-poly-Q models tend to be more severe with an earlier onset and therefore more difficult to recover from.

The overall picture seems exciting, but the details in the data are not as clear as the writing would suggest. In many of the figures and assays, they only look at mutant HD cells, and not fibroblasts from unaffected individuals,

It is difficult to quantify levels of Htt protein in wildtype fibroblasts because they express very low levels of protein and in general in this cell line the mutant protein does not accumulate much. Recently we found that HEK293 cells have quite high levels of wildtype Htt protein. We have produced a stable HEK293 cell line transfected with LC3-GFP-mCherry. We now have included in our manuscript results from this cell line of NCT-504’s effect on autophagosome formation, autophagy flux and Htt level in dose response and in a time dependent manner. Indeed modulation of autophagy and reduction of wildtype Htt levels can be observed in HEK293, indicating a biological effect upon inhibiting PIP4Kγ in a context outside of HD. Other potential effects of inhibiting PIP4Kγ on unaffected individuals are commented on by Lewis Cantley in their PIP4Kγ knockout mouse (and referenced in our paper) where the only observable effect is the activation of the immune system. Additionally we have included in Figure 4—figure supplement 1 data showing that decreasing PIP4K activity using a heterozygous loss of function (lof) or a neuronal expressed dominant negative (DN) does not cause neuronal dysfunction in *Drosophila* measured as motor performance.

and most of the readouts are only looking at one protein, while the proposed mechanism is to switch on autophagy in general. Autophagy, and its sub-categories of mitophagy, micro-autophagy, macro-autophagy, pexophagy are all dynamic system in a high rate of constant flux in the cell. If this is truly a drug lead for Huntington's disease, or any disease, they need to show some selectivity in the effected state and in the target. It seems that even in their loading controls, there is a massive global effect of shutting down PIP4Kγ either by SiRNA or global inhibitor. In other words, autophagy in all its forms is in dynamic flux for a reason, and just pushing it in one direction may be as problematic and inhibiting it.

As we note in the manuscript, numerous authors have described mutant Htt-dependent autophagy dysregulation in Huntington’s disease (HD). Especially important are several studies describing severe deficits in productive autophagy in HD cells and an inability to load mHtt and aggregates into autophagosome vesicles. Our data show that inhibition of PIP4Kγ increases productive autophagy of mHtt and aggregates. In the discussion we hypothesize that this is due to the observed shift in PI’s, which seem to promote the loading of mHtt and aggregates. In addition, this shift also upregulates autophagy flux, which might potentially increase the proteolytic rate of other ubiquitinated proteins tagged for elimination by autophagy, as well as damaged mitochondria (mitophagy is impaired in HD) and other cell damaged organelles. The impact of shifting autophagy dynamics can be insult-dependent and has to be evaluated in relevant disease models. Indeed upregulation of autophagy can prevent or promote apoptosis, depending of the cell model and its capacity to respond to stress. In previous KO mouse studies by Lewis Cantley, PIP4Kγ-dependent upregulation does not seem to have a major biological impact in viability or behavioral dysregulation, with no observable phenotype other than hyperactivation of some T cell populations, which it seem to be controllable by rapamycin. In the context of HD, levels of mHtt are elevated not only in striatal neurons, but also many other cell types, and numerous authors have described non-neuronal Huntington impairments, and therefore a global reduction of mHtt is perceived to be beneficial, as has been demonstrated with other autophagy modulators. Indeed our in vivo models indicate that this is the case, with an amelioration of the disease progression. Moreover this benefit of inhibiting PIP4Kγ is titrated by elimination of a single allele. Further dose dependent efficacy studies with a fully optimized PIP4Kγ inhibitor will be necessary to determine the adequate dose, schedule and potential side effects of this novel mechanism.

We have performed titration curves for NCT-504 in the Q45 patient fibroblasts. As shown in Figure 4—figure supplement 1, NCT-504 is not toxic until we reach the 10μM concentration, at which it does cause cell loss. We do see a significant effect on mHtt levels at 5μM. Thus we are able to reduce mHtt levels in non-toxic conditions.

We have also assessed the effect of the heterozygous PIP4K^LOF^ as well as the new PIP4K^DN^ allele on motor performance on their own in the absence of mHtt. As shown in the new Figure 5—figure supplement 1 neither allele causes significant behavioral improvement or worsening compared to the negative control as measured by climbing ability or speed

I have an conceptual concern that this data suggest that 97% of the huntingtin ORF is irrelevant for regulating the turnover of this protein by autophagic mechanisms, and it appears from this data that the molar stoichiometry is irrelevant, as whether they look at endogenous huntingtin proteins, or massively over-expressed small molecule fragments at orders of magnitude above physiological molarity, the compound works just as well to clear 'huntingtin', and likely everything else.

Our initial data focused in demonstrating that inhibition of PIP4Kγ reduces levels of mHtt and N-terminal-exon 1 aggregates in several cell models, and that the mechanism has a therapeutic benefit in a relevant N-terminal Htt-poly-Q *Drosophila* model of HD. As discussed above, now we have added new data using a FLQ200 *Drosophila* model showing similar results, with amelioration of the decline in motor performance. Our studies in the mechanism of action indicate an increase in autophagy flux, which is impaired both by expression of NT Htt-poly-Q and by expression of FL Htt-poly-Q. Our control proteins indicate that this mechanism does not clear everything else, but eliminate specific insults in Huntington cells.

My major concern is that I believe they have proven that PIP4K γ inhibition triggers a massive level of autophagy, but they have no data here that show any selectivity or health effects on the fibroblasts. Is this "curing dandruff by decapitation"?

We respectfully disagree with the interpretation that PIP4K γ inhibition triggers a massive level of autophagy. Instead PIP4K γ inhibition seems to elevate productive autophagy of mHtt and fragments, ameliorating disease progression.

Autophagy is a continuous and dynamic process carried out by cells to eliminate properly tagged proteins and organelles, by polyubiquitination (macrophagy, mitophagy) or KFERQ-like sequences (chaperone-mediated autophagy). The basal level of autophagy depends of stimulation by environmental stressors but also by intracellular factors such as genetic stressors, lysosomal homeostasis and level of energy production. Elevating the basal level of productive autophagy does not have to be necessarily detrimental for the cell in general and even less in HD, as productive autophagy is downregulated by the expression of mHtt and its fragments. Indeed, our siRNA experiments across cell lines show that elimination of the PIP4Kγ kinase activity does not induce cell death. This is further corroborated by the viability of full knockout animals with no growth or behavioral abnormalities. Furthermore any theoretical risk could be further modulated using the right dose and schedule of a PIP4Kγ small molecule inhibitor.

It is not clear for us how to assess the potential health benefit on fibroblasts, besides analyzing the reduction of the accumulated insult and the effect on cell viability. Therapeutically, any intervention is a balance between potential benefits and adverse effects which need to be carefully evaluated in relevant disease models (cell and animals if available) and eventually in human clinical trials. We believe that our data demonstrate the therapeutic benefit of this mechanism in relevant models of HD.

As detailed above, analysis of toxicity by dose titration in cells indicates that we are causing mHtt decrease at a non-toxic concentration. Furthermore decreasing *PIP4K* levels or activity in *Drosophila* does not cause any visible toxic effect in our assays. This is consistent with results in mouse models. Overall we believe the PIP4Kγ is a very safe target, and any toxicity that maybe observed with the NCT-504 is likely due to side effects not related to PIP4Kγ inhibition.

They reference a manuscript on a PIP4Kγ knockout mouse stating the investigators observed "no growth or behavioral abnormalities", except the manuscript actually describes a massive global hyperactivation of the immune systems in those knockouts, and the true validation of a drug target would be a conditional knockout. For what it's worth, Broad's Exac exomic sequencing database shows no homozygous null alleles of PIP4Kγ yet in humans. Thus, I have concerns that PIP4Kγ inhibition could really be a drug target.

Lewis Cantley et al. in their description of the phenotype of PIP4Kγ knockout mouse also mentioned a population of human subjects carrying polymorphisms at PIP4K2C locus linked with familial autoimmunity. Additionally these authors described the hyper-activation of the immune system in the null PIP4Kγ mouse, which they are able to correct using the mTORC1 inhibitor rapamycin, a well-accepted upregulator of autophagy initiation which is also able to ameliorate in vivo the HD phenotype, demonstrating that the autophagy-modulating effect of inhibiting PIP4Kγ is distinct from inhibition of mTOR. Importantly in our HD *Drosophila* models, we demonstrated a potential therapeutic benefit upon decreasing dPIP4K activity, indicating that a partial modulation of PIP4Kγ activity might be enough for observing a beneficial effect. Dose dependent efficacy studies with the HD animal models and eventually in HD patients with a fully optimized PIP4Kγ inhibitor will be necessary for determining the percentage of PIP4Kγ-inhibition and the schedule needed to minimize any potential immune side effects while maximizing the therapeutic benefit from mHtt elimination. Furthermore, the potential secondary effects of PIP4Kγ inhibitors such as a risk of hyper-activation of the immune system could be different in mHtt-carrying HD-patients (migration of primary microglia seems to be impaired by mHtt expression PMID: 27615381) that in wt HD-patients or in population carrying specific SNP’s at the PIP4K2C locus.

They do not state exactly how their HD fibroblasts are immortalized, but is seems that there is no difference in effect between HEK293, STHdHQ111, HD primary cells, or HD immortalized cells. This is rather surprising since the only known inducer of huntingtin expression is P53, and some of those lines have P53, and others do not. This has been a concern that has been ignored in most HD cell lines studies, despite over 80 manuscripts published consistently defining disrupted P53 biology in HD, from mouse to humans. So whether they have p53 or not, whether small fragments of full protein, everything gets cleared, in all lines.

To our knowledge all the cell lines used in our experiments, PC12, HEK293, MEF, non-immortalize and immortalize HD fibroblast, STHdHQ111, and mouse primary cortical neurons, are wt p53. For example:

HEK293: (https://www.ncbi.nlm.nih.gov/pubmed/0008504475).

PC12 cells: (https://www.ncbi.nlm.nih.gov/pubmed/16817227).

We have now indicated in the Materials and methods section that the fibroblasts were immortalized using SV40 large T antigen.

Major concernsFigure 1From the high throughput screen, they found NCT-504 and by the structure guessed that it was a kinase inhibitor, and tested this by kinomic screening, and used a cutoff of >65% inhibition to claim specificity. This is a very high cutoff, and the entire kinome screening data should be included a supplemental, not just 3/442 kinases. From my experience with pharma, even 20% inhibition is cause for concern on these screens. The IC50 is about 16μM for NCT-504. This is very high, and by their own data, the IC50 exceeds the toxicity threshold concentration in neurons.

We did include the entire kinome screening data as Table 1 – Source data 1, and apologize if it was not available by the reviewer somehow. It can be observed that at 10 μM NCT-504 only PIP4Kγ is robustly inhibited, with little or no effect in the rest of the 442 kinases in the panel, as might be expected for a non-competitive allosteric inhibitor.

Regarding the IC50 of NCT-504, we explained in the manuscript the differences in IC50’s obtained using several in vitro kinase assays. While the IC50 using isolated PIP4Kγ was 15.8 μM, the IC50 using DiscoverX binding assay was 350 nM. As a screening center, we are very familiar with these kind of discrepancies between biochemical assays, and in the manuscript refer to other authors describing similar problems with allosteric modulators. The main reason behind these discrepancies has to do with the assay conditions used for evaluating inhibition of isolated kinases, usually requiring extraordinary high levels of ATP in order to promote autophosphorylaton and activation and the need for using micelles or detergents to stabilize the active conformation of the kinase, something especially important for lipid kinases such as PIP4Kγ. In general, for these kind of kinases, the DiscoverX binding assay using *E. coli* or mammalian cell-expressed kinases labeled with DNA tag for qPCR readout, seems to more faithfully reproduce the activity of the kinase in its natural environment. Importantly, in order to properly evaluate the real PIP4Kγ inhibitory capacity of NCT-504 in a cell environment we measure its dose dependent modulation PI5P levels (and other PI’s), using siRNA as a control of the maximal elevation obtained upon elimination of the activity of PIP4Kγ. These measurements in cells clearly indicate that the concentration used in the cell assays to evaluate effects on mHtt and aggregates robustly inhibits PIP4Kγ and that blocking the activity of this kinase does not reduce cell viability. All the data reported in the original manuscript was carried out at non-toxic concentration of NCT-504 in the same cells. Now we have incorporated additional dose-response and viability in different cell lines and end points (PI’s levels, autophagy, mHtt levels, viability, etc.)

The concentrations of NCT-504 used for effects vary hugely across the figures. In Figure 1, and from text, IC50 is quoted at 15.8μM. In Figure 1—figure supplement 1, Figure 1—figure supplement 2μM is used to reduce aggregation by 50% effect, so whatever reason they see effect, it seems to be unlikely through PIP4Kγ inhibition.

The basal level of autophagy is dynamic process and varies in an insult dependent manner (some cell lines expressing poly-Q’s accumulates more mHtt or aggregates than others), in a cell type dependent manner (genetic background affects the basal level of autophagy), upon assay conditions (confluency, passage, nutrient level), time (we evaluate modulations of PI’s upon exposure to NCT-504 in a time dependency manner) and of course in a concentration dependent manner. We previously indicated that the concentration of NCT-504 able to induce a robust effect is cell type dependent, as would be expected for a modulator of autophagy. Now we present the data using the same concentration across several cell lines (Figure 4) or dose-dependent in a single cell line (Figure 3—figure supplement 2).

Figure 2: concentration of NCT-504 is now 10μM, less than the IC50, and only one dose tested.

We include now PI-dose titration (Figure 2—figure supplement 2) and cell viability (Figure 2—figure supplement 1) in MEF with NCT-504. We use in these cell 10μm as we want to observed a robust effect on PI’s modulation at a non-toxic dose for comparison with the shRNA data. As we mention before the extent of the PIP4Kγ inhibitory effect is cell dependent, as different cells have level of basal autophagy and impediments (or not) in autophagy flux. Upon evaluation of a specific cell line, we select the adequate dose depending, among other things, of the impact of NCT-504 on cell viability, being sure that we report the effect a non-toxic doses.

The loading controls seem to switch between α-tubulin, actin and GAPDH. For these studies looking at autophagic flux, as in Figure 4 they need better controls, not an insoluble polymer of actin. I'd like to see GAPDH and at least two inner mitochondrial proteins to show me the cells are not just completely autophagic or apoptotic. Images are essential.Again, for Figure, 4, now 2μM of NCT-504 is used again in Q68 cells, but then 10μM in Q45 cells. This is plus or minus 500%.Figure 4: they need to show the whole blot in a supplemental figure. I see a severe hit on actin levels in Figure 4 with PIP4K2C_si, and in Figure 4 with 10μM NCT-504, which concerns me that these cells might be heavily auto-degrading or just dead.

As suggested by the reviewer, we have included GAPDH in our dose-dependent experiments using patient fibroblast. We also have included now the effect of NCT-504 treatment on cell viability of numerous cell lines. Impaired mitophagy has been reported to contribute to HD (PMID: 26268247) and therefore it is possible that the increment in productive autophagy by NCT-504 might also affects the levels of mitochondrial proteins, within being necessary a bad outcome. For example we do not see an effect on cell viability upon silencing PIP4Kγ, and definitively we observe a beneficial therapeutic effect in *Drosophila*. Additional studies to better characterize these effect in cells and the effector proteins are necessary but we believe that are out of the scope of these paper.

In Figure 4—figure supplement 4, they show a very unconventional assay for neuronal toxicity, in that they are no showing any images of neurons or other measure of health, or even life, but their own data shows a pretty severe "toxicity" at 10μM NCT-504, the exact concentration used in HD fibroblasts, and still below their calculated IC50.

The effect of NCT-504 in Htt aggregates and reactive species evaluated in mouse primary cortical neurons transduced with 72Q shown in Figure 4—figure supplement 3 is a 5μM and 2.5μM, The same concentrations were used in HD fibroblasts and striatal cells from knock-in HD mice (Figure 4) without major impact in viability.

Figure 4—figure supplement 4, that western blot is not appropriate data to conclude anything.

We have eliminated Figure 4—figure supplement 4, leaving only 4A and 4C.

Figure 5. They now switched back to a severe allele Q128 (extremely rare) and only small fragment 1-231. Why did they not use full huntingtin?

Please see answers regarding additional *Drosophila* data above.

Reviewer #3:1) How was NCT-504 selected for these studies from published the high-throughput screen? Was this best hit?

NCT-504 was produced upon medicinal chemistry optimization toward potency and metabolic stability of the screening hit ML168 (reference 33). The manuscript has been corrected to clarify this question. Details of the structure-activity relationship will be published in a following manuscript and we consider that topic out of the scope for this paper which focuses on the validation of PIP4Kgamma as a druggable target for Huntington Disease.

Because of the complexity of these pathways, it would be helpful to have a schematic early on to show the various components of the pathway, how each of the PIP derivatives are generated and where this kinase fits into general scheme.

We already have a large amount of figures in this manuscript and we ask the editor for guidance in addressing this question. We can include such as scheme, similar to that found in references 32 and 42.

2) Figure 4 - Why not show more complete set of westerns as in C and D? Need quantification for Figure 4.

To do quantification in patient fibroblast in dose response is quite challenging because the overall levels of Htt protein are low and NCT-504 seems to impact cell viability of these cell lines at double digit μM concentrations. Alternatively we have observed high levels of Htt in HEK293 cells and in these cells NCT-504 des not impact viability at all the tested doses (Figure 3—figure supplement 2) that allow us to use a FRET assay to quantify Htt levels in dose response.

3) Do the fibroblasts have HMW species? This is not mentioned and if they are present, why they are not included should be addressed.

As far as we know there are no HMW species in the fibroblast lines. We do not see any HTT left in the stacking portion when running WB gels. In addition, when we have used oligomer specific TR-FRET analysis we have not detected any signal in these cells.

4) A potentially important experiment would be using a pulse chase or other such assay to measure whether this truly increases degradation of the protein.

We have included now the effect of NCT-504 in neurons on autophagic flux determined after 24 hours using a Dendra2-LC3 photoconverted reporter (Figure 3—figure supplement 3) and we have titrated Htt levels along autophagosome formation and autophagy flux in HEK293 cells (Figure 3—figure supplement 2).

5) Please describe length of HTT fragment in the 128Q HTT transgene in flies.

This has been added to the Materials and methods.

6) A citation should be included for the Neuron paper from the Davidson lab when discussing Rhes and Rheb.

We have included the requested citation.

7) It is a stretch to say that compound modulates HTT proteolysis as there is a single Western blot with one panel.

We have include now other methods of quantification of Htt levels using a FRET assay. The western bolt analysis was carried out by triplicated in each cell line and we used several types of cell lines.

[Editors' note: the author responses to the re-review follow.]

Essential revisionsIt is critical that the authors pay close attention to and respond accordingly to reviewers 1 and 2 concerns that the strength of this manuscript is decreased considerably by the style/manner in which it is written. Extensive editing to the text and figures are needed so that the manuscript flows smoother and rationale for the studies are clearer.

We read the decision letter carefully and have addressed every concern. We have extensively edited the writing style/manner without changing the content or essence of the paper. The paper was reviewed and modified by several authors and colleagues who speak English as their native language. Almost all figures have been edited to have consistent font and better quality. We have also included concentrations of compounds in all figures.

Reviewer #1:In this new submission of a previously submitted manuscript the ability of a small molecule to inhibit PIP4Kγ and impact HD-like phenotypes in cellular and animal models expressing mutant Htt (full-length and fragment) is presented. Overall while this work addresses a very interesting issue and additional data using a FL-Htt fly model have been added, it remains difficult to read/follow in several key places.

We have taken this criticism about the writing style seriously and changed the text in most places. While the basic content and essence of the paper remains unchanged the writing has been modified extensively. We believe the narration and discussion is much improved. Please let us know if you have any additional questions

Specific points:Overall the manuscript remains hard to read, needing English editing in many places. In several places statements are referenced when they should be. For example, second paragraph of Results – "Similar discrepancies in potency […]" needs to be referenced.

This has been addressed by the addition of two new references:

Rudolf, A.F., et al., A comparison of protein kinases inhibitor screening methods using both enzymatic activity and binding affinity determination. PLoS One, 2014. 9(6): p. e98800.

Smyth, L.A. and I. Collins, Measuring and interpreting the selectivity of protein kinase inhibitors. J Chem Biol, 2009. 2(3): p. 131-51

Clarity of Figure 3 would be improved by including concentrations of compounds used within body of the figure and not just in the legend.

We have addressed this. We have included concentrations and added clear labels with a consistent Arial font throughout the paper.

Reviewer #2:This is a revised manuscript and the authors were very responsive to the reviewers.The work described identifies PIP4Kγ as a therapeutic target for HD and shows that pharmacological inhibition of the enzyme promotes mutant HTT degradation and autophagy. They provide in vivo evidence that lower the levels of PIP4Kγ is beneficial in a fly model of HD-retinal assays and motor performance. This is a promising direction to pursue as a possible therapy for HD and is in line with current treatments directed at lowering mHTT protein levels. Generally the quality of the data is strong. Minor points are that there are a number of formatting typos. Also the figures are not very consistent. Formatting is varied-different fonts, different graphing and repetitive information in legend and graphs.

We appreciate the reviewer’s support.

Our work was a collaboration between multiple institutes and we are proud that we were able to work together and bring different expertise to the table to discover to validate a new target for Huntington’s disease. However, as the work was done in different sites the data for the figures was compiled with various format/fonts/styles. We understand that the figures needed to be consistent and have addressed this by changing, wherever possible, the figures’ text to a consistent Arial font throughout the manuscript.

Reviewer #3:This manuscript describes the effect of inhibition of the PIP4Kγ in models of Huntington's disease. There is extensive description of the effect of the drug on lipid metabolism and autophagy. There are also data from cell models including HD fibroblasts. The reduction of Htt levels is of interest. However effects on Htt aggregates are difficult to interpret since aggregates may be toxic or protective. There is also an interesting effect on an HD Drosophila model. It would be beneficial if the investigators had some more substantial data on models more directly relevant to HD pathogenesis. For instance instead of using HD fibroblasts it would be beneficial to use HD iPS cell models. In addition, there are no data from HD mouse models. The processes of autophagy are cell type specific, and may be different in Drosophila and mammalian systems, so in vivo data from mice rather than flies would be advantageous.

These comments, while very relevant, are outside the scope of the paper focused in the validation of PIP4kγ as a target for Huntington’s disease. We show in multiple cell-based assays, including patient fibroblast and stratial neurons, that inhibition of this lipid kinase via si or sh RNA leads to a reduction of mutant or wild-type Htt protein. In addition, we disclose a tool compound that acts as an inhibitor of PIP4kγ and is able to correct the disease phenotype in multiple cell lines.

Our team does not have access to iPS cells at the present moment. Moreover striatal neurons differentiated from Htt iPSC’s do not present aggregates, and therefore we would be able only to use them for measuring total levels of mHtt and autophagy dynamics. We already have measured these events with well accepted mammalian cellular models. Thus, using compound treatment we show the following in multiple different cell lines (included in bold are cells of neuronal origin):

1) Reduction of GFP-mHttQ103 in PC12 cells, a neuroblastic cell line of rat adrenal phaeochromocytoma origin (Figure 1).

2) Reduction of wt Htt from 293A cells which is derived from primary embryonal human kidney cells (Figure 3—figure supplement 1)

3) Reduction of mHtt protein levels in an HD patient fibroblast cell line (Q68), Figure 4

4) Reduction of mHtt protein levels in an HD patient fibroblast cell line (Q45), Figure 4

5) Reduction of mHtt protein levels in immortalized striatal cells from knock-in HD mice (STHdhQ111), Figure 4.

6) Reduction of accumulated HTT-exon1 aggregates in HEK293T cells transfected with GFP-HTT(exon1)-Q23 or GFP-HTT(exon1)-Q74, Figure 1—figure supplement 1.

7) Reduction of wt Htt in 293A cells (derived from HEK cells), Figure 3—figure supplement 1.

8) Reduction of GFP-Htt(exon1)-Q74 aggregates in Atg7+/+ MEF cells (Figure 3—figure supplement 3)

9) Reduction of mt Htt in mouse primary cortical neurons transduced with Htt(exon1)-Q72 lentivirus, Figure 4—figure supplement 4

The above data strongly indicates our approach of inhibiting PIP4Kγ to upregulate autophagy and reduce Htt will work across multiple cell line including those of neuronal origin, and therefore is not cell specific.

NCT-504 is a tool compound and needs to be optimized further towards experiments in animal models. We have evaluated the pharmacokinetics of the compound and while it has decent plasma exposure it fails to cross the blood brain barrier. We intend to improve the ADME (Absorption, Distribution, Metabolism, Elimination) properties of the compound via the generation of new analogues.